# Collective search in ants: Movement determines footprints, and footprints influence movement

**Stefan Popp** ⃝*, **Anna Dornhaus**

Department of Ecology and Evolutionary Biology, University of Arizona, Tucson, Arizona, United States of America

* popp@arizona.edu

**Data Availability Statement:** All raw and manipulated ant track files are available from the Open Science Foundation database (link: https://osf.io/v65tj/?view_only=350bd527187b4d829458cda199942bb0).

## Abstract

Collectively searching animals might be expected to coordinate with their groupmates to cover ground more evenly or efficiently than uncoordinated groups. Communication can lead to coordination in many ways. Previous work in ants suggests that chemical 'footprints', left behind by individuals as they walk, might serve this function by modulating the movement patterns of following ants. Here, we test this hypothesis by considering the two predictions that, first, ants may turn away from sites with higher footprint concentrations (klinotaxis), or, second, that they may change their turning patterns depending on the presence of footprints (klinokinesis). We tracked 5 whole colonies of *Temnothorax rugatulus* ants in a large arena over 5h. We approximated the footprint concentration by summing ant visitations for each point in the arena and calculated the speed and local path straightness for each point of the ant trajectories. We counterintuitively find that ants walk slightly faster and straighter in areas with fewer footprints. This is partially explained by the effect that ants who start out from the nest walking straighter move on average further away from the nest, where there are naturally fewer footprints, leading to an apparent relationship between footprint density and straightness However, ants walk slightly faster and straighter off footprints even when controlling for this effect. We tested for klinotaxis by calculating the footprint concentrations perceived by the left and right antennae of ants and found no evidence for a turning-away (nor turning-towards) behavior. Instead, we found noticeable effects of environmental idiosyncrasies on the behavior of ants which are likely to overpower any reactions to pheromones. Our results indicate that search density around an ant colony is affected by several independent processes, including individual differences in movement pattern, local spatial heterogeneities, and ants' reactions to chemical footprints. The multitude of effects illustrates that non-communicative coordination, individual biases and interactions with the environment might have a greater impact on group search efficiency and exploratory movements than pheromone communication.

**Funding:** AD was funded by National Science Foundation grants IOS-1455983 and DBI 1564521 and Defense Advanced Research Projects Agency SBIR 4024140. https://nsf.gov/ https://www.darpa.mil/ The funders had no role in study design, data collection and analysis, decision to publish, or preparation of the manuscript.

**Competing interests:** The authors have declared that no competing interests exist.

## Introduction

Foraging organisms face the general problem of how to search in space to discover new resources. In central place foragers, who return to a fixed base like a nest between exploration trips, long-term search efficiency is mostly determined by how much individuals avoid already searched areas, balanced with how close they can stay to the nest to minimize return travel. One way of doing so, for example, is a regular meandering pattern [1]. For colonies of central place foragers like social insects, the success of individuals is additionally tied to that of the whole colony. Here, searchers need to also avoid areas where nestmates have recently searched and presumably depleted any resources. There is empirical evidence for colony-wide search coordination in several ant species, which might partition the search area among individuals, or the type of resource (carbohydrates vs. proteins) [2–4]. Additionally, colonies of the ant *Temnothorax albipennis* appear to cover the area around the nest more evenly than the sum of individuals would [5].

### Central place search efficiency and chemical cues

The research of Hunt et al. [5] hypothesized that ants can achieve this by avoiding chemical footprint pheromones. These are passively deposited by insects and consist of hydrocarbons likely originating mostly from the cuticle [6–9]. There are two ways in which such footprint cues might be used by ants to increase colony search efficiency by avoiding densely searched areas. Firstly, footprints could simply be an aversive stimulus to searching ants, making them turn away from already walked-on areas (negative klinotaxis). There is direct evidence from manipulation studies that some ant and bee species avoid conspecifics' or their own footprints [9–12]. Secondly, ants might change the walking speed and/or straightness of their walking path while on footprints. This 'klinokinesis' can also lead to a quick escape from already-searched areas [13]; its effect is similar to the foraging strategy 'win-stay, lose-shift' in biasing where time is spent [14]. Bacteria use 'klinokinesis' to swim into or out of gradients [15] and *Caenorhabdidis elegans* worms use both mechanisms together [16]. There are hints that *Lasius niger* ants turn more when walking off footprints, although only the frequency of U-turns was analyzed in those studies [17, 18]. Bumble bees, however, likely learn whether footprints are indicative of resources or the absence thereof [19], making it plausible that ants might also use memory of perceived likelihood of encountering resources to change their behavioral responses to chemical cues.

### Testing mechanisms of coordinating search

Here we ask how ants may achieve more efficient area coverage through coordination with nestmates. We test the two hypotheses of klinokinesis and klinotaxis introduced above. We specifically measure changes in path straightness of all extranidal ants for a full colony with unrestricted access to a large arena, which constitutes a more naturalistic setting than in previous work on this topic. We quantify in detail the spatial pattern of presumed footprint pheromone deposition around the nest, and test how ants change their walking speed and turning behavior in reaction to footprints and the distance to the nest. This allows us to quantify graded responses to different realistic footprint concentrations as well as detect other aspects of colony-level coordination, such as the roles of different individuals, which turns out to be an important aspect of colony search strategy. Previous studies on ants only considered U-turning frequency and/or walking speed, not dynamically changing path straightness, and used binary comparisons of behavior on and off footprints, without regard to a possible graded response as well as the pattern in which footprint concentrations are likely to be encountered around the nest. While previous studies [17, 18, 20, 21] controlled well for interactions with

nestmates by only allowing one individual on the test setup, this may have limited insights into how ants coordinate their search movements when exiting and entering the nest freely over long periods of time.

## Study species

We used colonies of the ant *Temnothorax rugatulus*, a species that is in the same genus as *T. albipennis*, which was used in one of the previous studies [5]. Both species are presumed to be scavenging or hunting for microarthropods, and to achieve most of their resource intake through solitary foragers which search for new resources in every bout, while not using mass recruitment pheromone trails [22–24]. These ants also experience high selective pressure to find new nest sites [25–27] and competing or potentially invading ants [28, 29]. Searching ants may thus be looking either for food, nest sites, or competitors. Their foraging range can extend to more than 10m in radius [22]. This solitary, repeated search for small items puts a high selective pressure on the efficiency of the search strategy in these ants, in comparison to mass-recruiting ant species, which rely heavily on recruitment efficiency. *T. rugatulus* employs a meandering search pattern, covering an area more densely while visiting the same areas less often than alternative strategies [1]. This genus uses visual landmarks for navigation and orientation [30–34], and *T. rugatulus* changes their exploratory movements with increasing familiarity with the environment (*in review*, *will be updated once either paper will be published*). Even though foragers in this species are thought to not be attracted to or follow the trails of nestmates toward food [31], chemical markings from other ants are still important cues for colony coordination. When a colony emigrates to a new nest site, they can use existing chemical markings to move closer to a food-rich area [35], and individual foragers can discriminately use their own footprints as navigational guides and even to estimate internal nest area, especially in the absence of other information [31, 33, 36].

## Methods and results

### General setup

We largely used the materials and methodological procedures described in detail in Popp & Dornhaus 2023 [1], with some changes outlined below. In short, we collected five colonies of *Temnothorax rugatulus* ants 5 months before the experiments from the wild. We filmed them for 5 hours each in an empty 2x3 m arena, while all ants of the respective colony had unrestrained access to the arena. We then tracked ants using TRex [37] and corrected tracks, partially manually, and partially with custom MATLAB routines. In contrast to Popp and Dornhaus 2023 [38], we only included data of the third day of ant exploration for this analysis, where we had installed a clean paper floor covering, ensuring that ants are generally familiar with the surroundings but are not exposed to footprints from prior days. Points less than ca. 5 cm from the walls or an apparently repellent tape strip on the underside of the top paper layer were excluded, as well as tracks shorter than 30 cm, eliminating wall-following behavior from our data. In the end, the dataset included 3382 tracks coming from ca. 200 ants, due to multiple tracks corresponding to the same ants. To avoid spurious angles resulting from tracking imprecision of still or stopping ants without using arbitrary thresholding, tracks were resampled from 25fps to an equidistant regime, such that each ant track consists of a set of discrete points, where each point along the walked path is 2 mm away from the previous point. Resampling high-frequency movement tracks is an important step for analyzing the data on the biologically most meaningful scale [39]. All analysis code was written in MATLAB R2021a (The MathWorks Inc., Natick, MA, USA) and can be found in OSF (https://osf.io/v65tj/?view_only=350bd527187b4d829458cda199942bb0).

## Calculating footprint concentration

We assume that ants passively deposit chemical footprints during walking [7], and that these footprints accumulate with repeated ant visits. We calculated a hypothesized 'concentration' of footprint pheromone an ant encountered at every point on her movement track by summing the number of ant points that have been made in a radius of 3.5 mm around the current focal ant point, based on the video up to that time. This corresponds reasonably well to the number of ants having passed through the antennal radius of the focal ant.

The chemical footprints of ants are generally composed of compounds of different volatility [8], making it possible that ants only react to freshly made markings. We thus only include those points in the footprint concentration calculation which were made less than 1h before the ant visited this spot. We chose this cutoff based on estimations in other ant species [7, 8, 40]. We refer to this time as 'evaporation time'. Even though ants may still detect (parts of) the footprints made earlier, they might reduce or change their reactions to them. Our method thus assumes constant rates of deposition per distance walked, linear accumulation, and negligible diffusion of footprint chemicals up to the 'evaporation time'. It is difficult to estimate the impact of these approximations due to the scarcity of information about the chemical properties of footprint chemicals. However, to ensure robustness of our results to our estimated evaporation time, we repeated our analyses with 'evaporation times' of 5 min and 5h (S2 & S3 Files, respectively).

## Straightness calculation

Straightness of the ants' movement was calculated for each point (= focal point) as the Euclidean distance between the 10th point preceding and the 10th point following the focal point, divided by the total walked distance between these two points (the walked distance is always 40 mm since points are 2 mm apart). Straightness values thus lie between 0 (start and end point coincide, i.e., path is looping) and 1 (Euclidean distance approximates walking distance, i.e., the path is a straight line). An example track with changing straightness is depicted in Fig S1.1 in S1 File. We chose the 40 cm window to reflect the path straightness on the scale of the meandering behavior reported in Popp & Dornhaus 2023 [38].

## Statistical analysis

We used (generalized) linear mixed models for our analyses. When data were pooled across ants of all trials, we used 'colony' as a random factor. For any straightness analyses, we used the binomial distribution family due to straightness values being bounded between 0 and 1. When computing within-track correlations, we used (generalized) linear models instead. All statistical analysis were carried out in MATLAB R2021a (The MathWorks Inc., Natick, MA, USA).

## Results overview

We see ants moving straighter and faster when walking across fewer assumed footprints (Figs 1 & 2). We show that this is driven by at least 3 effects, summarized here but described in detail in the next sections. 1) A self-selection effect based on individual differences between ants, where i) there are fewer footprint markings farther away from the nest (Fig 2a and 2b), and ii) ants who consistently move straighter move farther away from the nest (Fig 4a). Ergo, straighter moving ants will be on low footprint concentrations. However, even when points are binned by their distance to the nest, ants walk straighter and mostly faster on lower footprint concentrations (Fig 5). 2) In areas where ants walk less straight due to factors other than

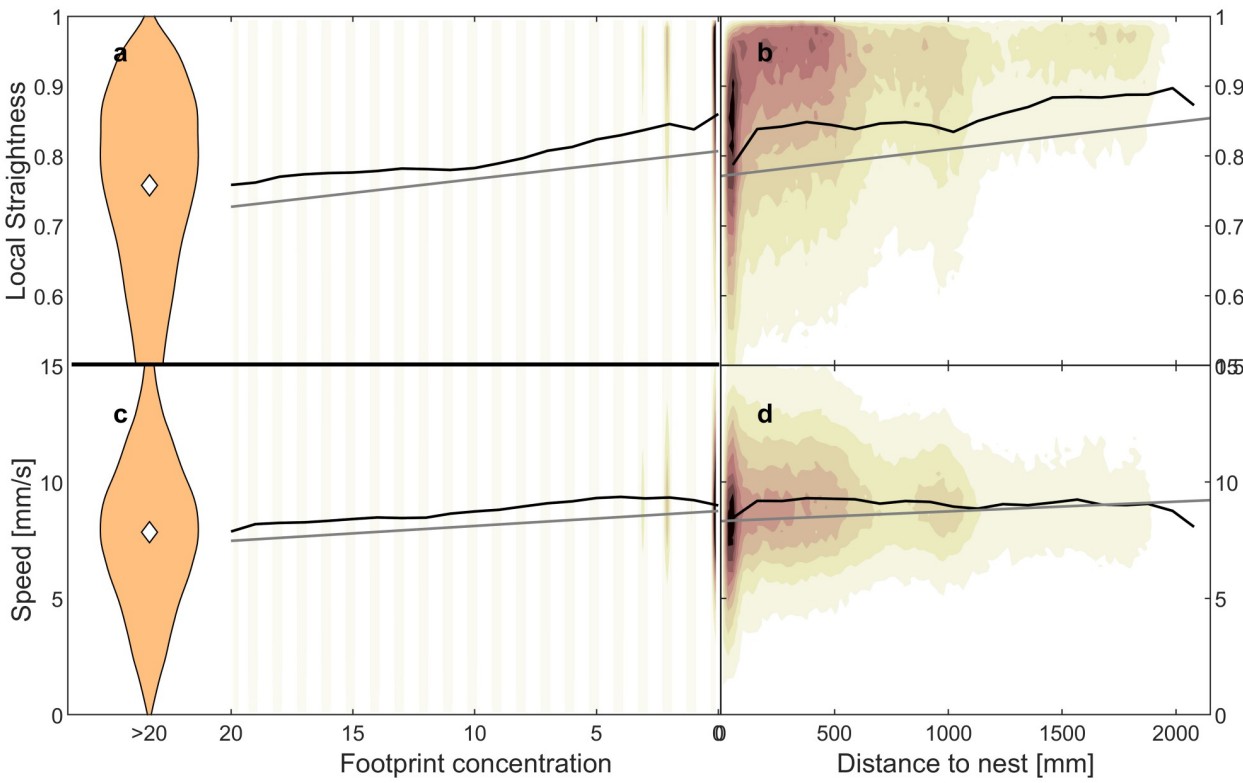

**Fig 1. Colony-level reactions to footprints.** a), c) Straightness and speed increase with decreasing footprint concentration. b), d) Straightness and speed increase with distance to the nest. Black lines are medians per footprint and nest distance bins, respectively. Gray lines are regressions from the LMMs. Faint beige to black areas are kernel density estimations of raw points. Violin plots show binned data for all footprint concentrations greater than 20, with the white diamond being the median. Statistical tests were performed on unbinned data. Note that the x-axis of the footprint concentration panels are reversed to be consistent with the 'Distance to the nest' plots, as the highest footprint concentrations are around the nest (i.e., towards the left side on both graphs). The y-axis limits are set to clearly show the slope, while cutting off only small parts of the tails of the data distribution.

footprints, there will be a higher footprint concentration (Fig 6a). 3) Controlling for the first two effects, ants still walk less straight on footprints (Fig 7a), thus supporting the klinokinesis hypothesis. We do not find evidence that ants turn into or away from increasing footprint concentration gradients (Fig 8), rejecting the klinotaxis hypothesis.

### Ants move straighter and slower further from the nest

When pooling points between all trials, ant movements are significantly straighter and faster the lower the footprint concentrations are they are currently walking on (Fig 1a and 1c; e.g., 12% straighter and 11% faster on 20 footprints compared to 0 footprints), and the farther they move from the nest (Fig 1b and 1d; e.g., from 7.9 straightness and 8.5 mm/s speed near the nest to 8.9 straightness and 9.5 mm/s at 1.5 m away from the nest; see Table 1 for statistics). As expected, high speeds are correlated with straight movements (Fig S1.3a in S1 File).

These results are qualitatively the same when considering individual tracks: more tracks (32.7%) are less straight on more footprints (negative correlation, left, darker gray bars in Fig 2a); only 16.9% of tracks are significantly straighter on more footprints (right, lighter gray bars in Fig 2a; LMMs of straightness~footprints). Similarly, more tracks (29.4%) are slower on more footprints, while only 2.1% of tracks are faster with increasing footprint concentration (Fig 2b; LMMs of speed~footprints).

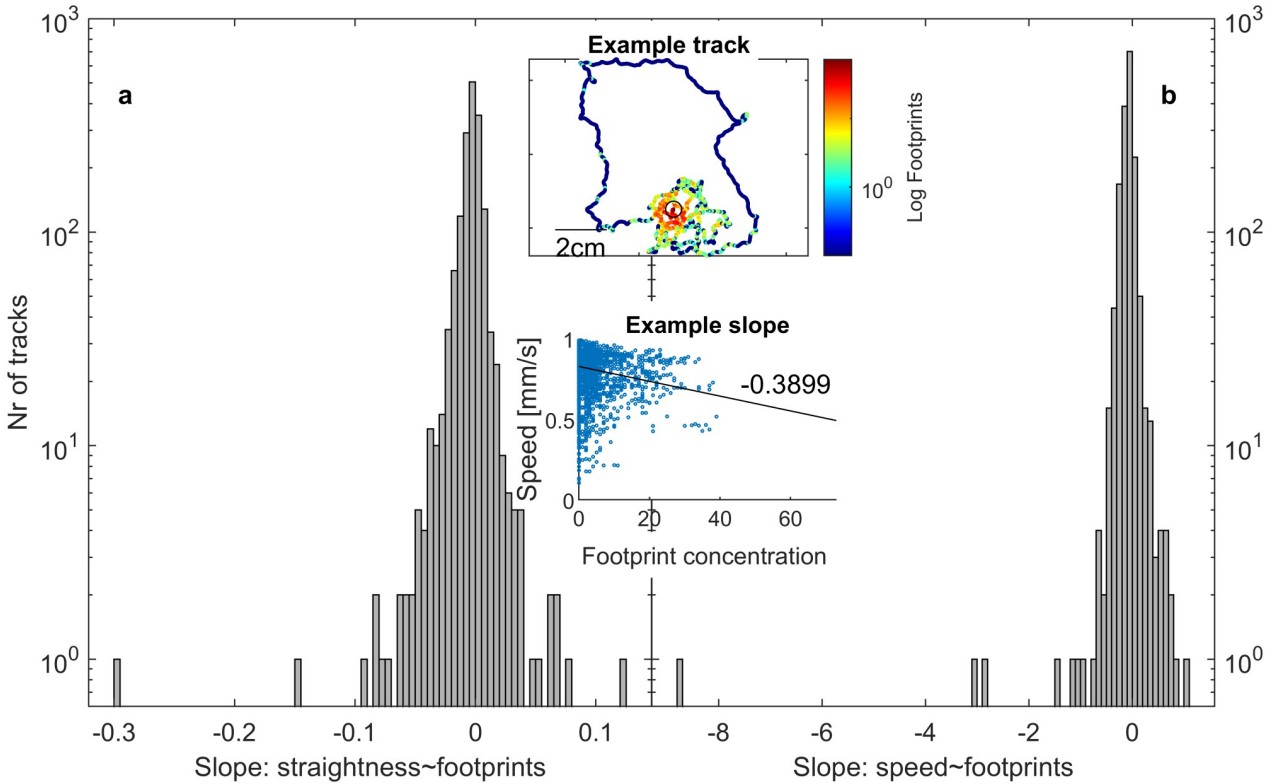

**Fig 2. Reactions to footprints by tracks.** The majority of ants walk a) straighter and b) faster on lower footprint concentrations. Histograms are of the slopes (for each track) of the linear models of straightness or speed ~ footprint concentration. Negative values (darker gray bars) mean ants are walking less straight or slower with increasing footprint concentration. Top inset: example track, colored by the footprint concentration the ant is currently walking over. Black open circle indicates the nest location. Bottom inset: scatterplot with slope of the linear model of straightness~footprints for that track.

## Straighter moving ants walk on lower footprint concentrations

A parsimonious explanation of why ants walk straighter and faster on lower footprint concentrations is that ants are not reacting to footprints, but rather that straighter tracks reach farther away from the nest, where there are also many fewer footprints (Fig 3a and 3b; Fig S1.3 in S1 File). Specifically, there are very high concentrations around the nest in a radius of about 10 cm, while the rest of the arena is fairly evenly walked on (Fig 3b).

We tested this hypothesis of consistent differences between tracks by evaluating whether the mean straightness of an ant's movement near the nest (<10cm) predicted that ant's average distance from the nest across the rest of her track.

**Table 1. Statistics of the straightness and speed versus footprints and nest distance comparisons.**

| Name | Estimate | SE | t | DF | p | Lower | Upper |
|---|---|---|---|---|---|---|---|
| straightness~footprints | -3.98e-3 | 2.07e-5 | -192 | 1.78e6 | <0.01 | -4.02e-3 | -3.94e-3 |
| speed~footprints | -6.4e-2 | 3.49e-4 | -183 | 1.81e6 | <0.01 | -6.47e-2 | -6.34e-2 |
| straightness~nest distance | 3.85e-5 | 2.33e-7 | 165 | 1.78e6 | <0.01 | 3.81e-5 | 3.9e-5 |
| speed~nest distance | 4.19e-4 | 4.22e-6 | 99.2 | 1.81e6 | <0.01 | 4.1e-4 | 4.27e-4 |

All models are LMMs with 'colony' as a random effect. Corresponds to Fig 1.

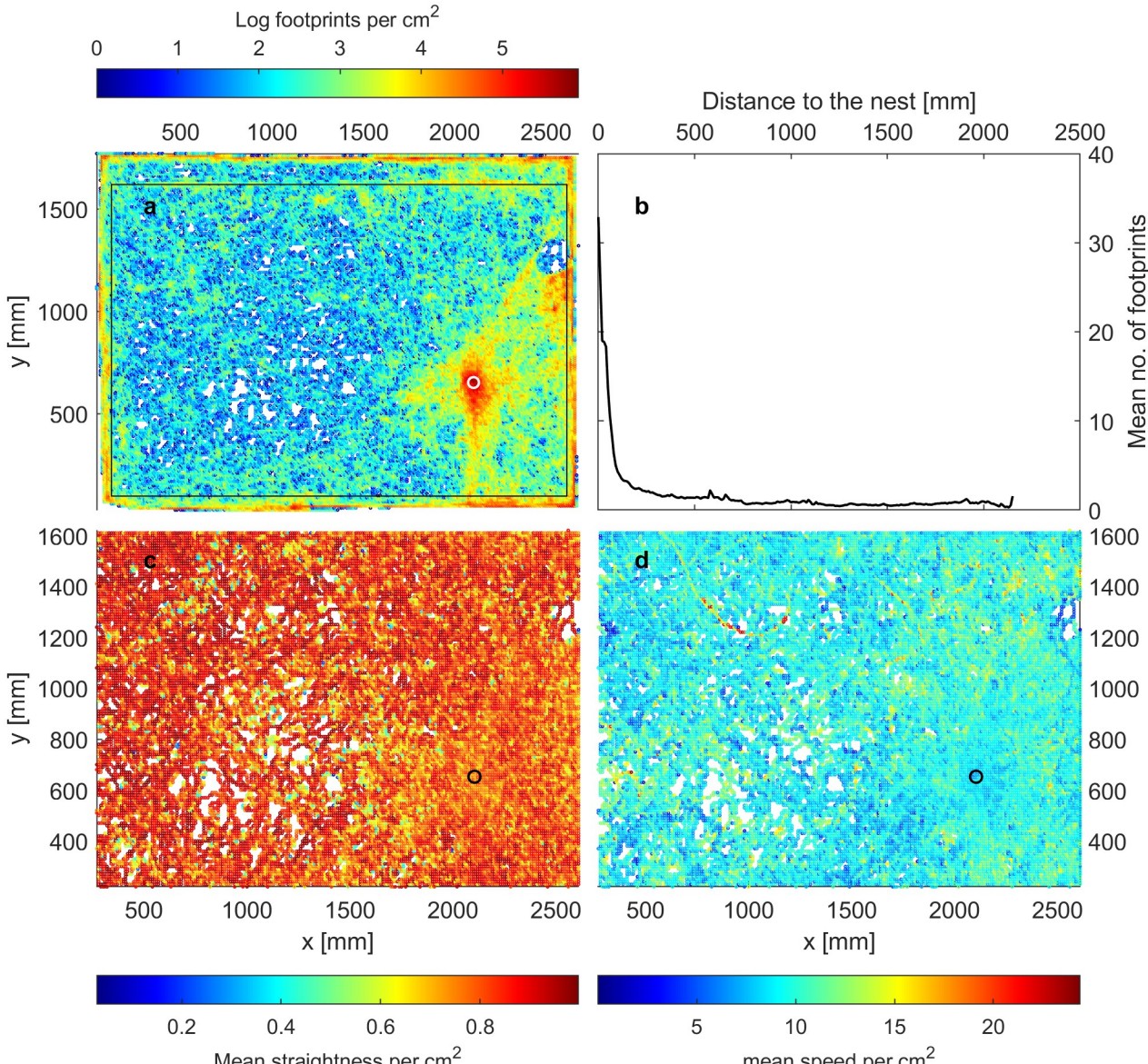

**Fig 3. Spatial distribution of footprints and movement characteristics of all 5 colonies pooled.** a) Heatmap of ant visitation, binned into pixels of size 1 cm. Points outside the black rectangle were omitted from analysis. Only data of the last hour of the experiment are included in this graph. Note that ants at different timepoints experience a slightly different footprint landscape. b) Mean footprint concentration by distance to the nest. c) & d) heatmaps of mean straightness and speed per pixel, respectively. Only the inner area outlined in a) is shown here. In a), c) & d) white and black circles indicate the nest location.

We indeed found that ants that start out straighter near the nest moved farther away from it, by roughly 74 mm per 0.1 straightness (LM mean(distance)~straightness_near_nest: Estimate: 741, SE = 79.1, t = 9.38, p < .001; Fig 4a). We found the opposite effect on speed: ants that are slower near the nest move farther, by roughly 12 mm per mm/s speed (LM mean(distance)~speed_near_nest: Estimate: -12.1, SE = 3.39, t = -3.56, p < 0.001; Fig 4b).

Within individual tracks, ants tend to increase their straightness and speed with increasing distance to the nest (LMM with track ID as random factor: straightness: SE = 52.02, t = 1.78e6, p < 0.001; speed: SE = 5.73e-6, t = 66.45, p < 0.001). Note that for speed, the intra-track effect

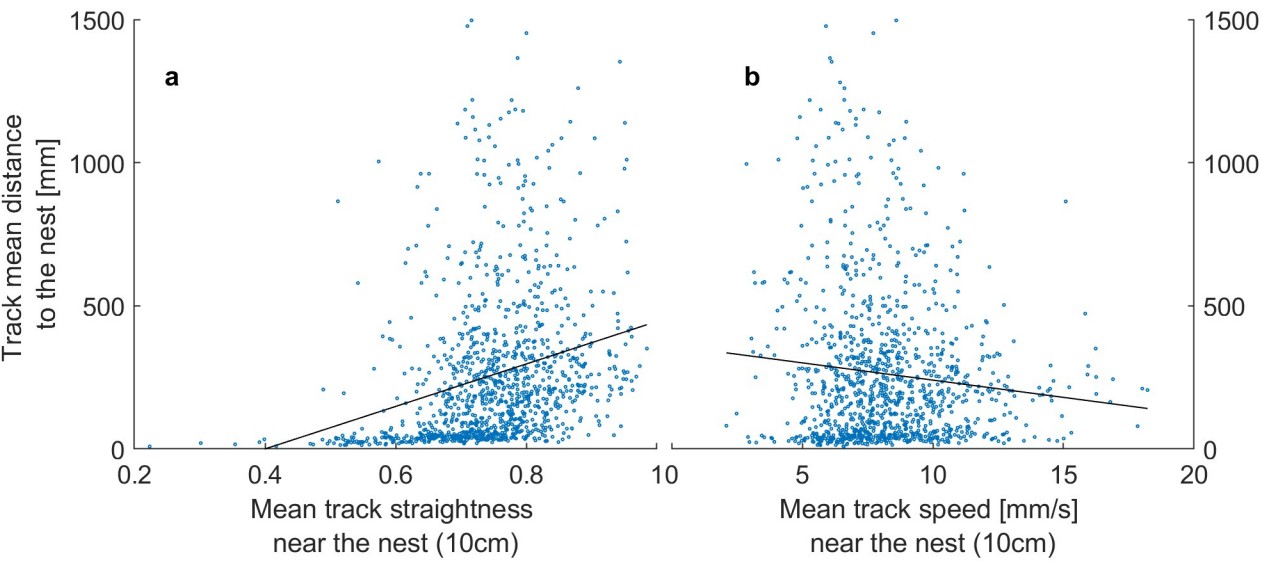

**Fig 4. Initial movement influences how far an ant disperses from the nest.** a) Tracks which start out straighter move on average farther away from the nest (where there are also fewer footprints, Fig 3b), but b) tracks which are faster near the nest move less far from the nest than those which are slower near the nest. Black lines are slopes of linear models.

(faster as the ant moves away from nest) is in the opposite direction to the inter-track effect (slower ants move farther away), leading to difficulties in interpretability of the overall analysis (Simpson's paradox). These results confirm that the high straightness of ants far away from the nest is due to both inter- and intraindividual correlations.

### Mostly lower straightness and speed on higher footprint concentration irrespective of distance to the nest

If the distance to the nest was the only factor determining the correlation between straightness or speed and footprint concentration, we should see no such correlation if we control for the distance to the nest. We thus binned the points by the distance to the nest into 20 bins, such that each bin contains a roughly equal number of points. We find that even within bins ants walk straighter and mostly faster on lower footprint concentrations by up to 0.15 straightness points and 1.2 mm/s walking speed per 10 footprints, suggesting an additional effect of footprints on the movement patterns of ants (Fig 5; Table S1.4 in S1 File for stats). The speed correlation is only non-significant or positive between roughly 45 and 70 cm (Fig 5b; Table S1.5 in S1 File for stats).

### Some areas make ants walk less straight, and thus deposit more pheromones there

So far, we assumed that the correlation between footprint concentration and movement behavior reflects a causation from footprints to behavior. However, it is also possible that there is a causality opposite to this assumption. If there are attractive areas where ants walk tortuously (for reasons unrelated to footprints), this will also make them deposit a lot of footprints in that small area. To test this possibility, we first binned the data into 2x2 mm pixels and then calculated, separately for all distance bins, the correlation between the straightness of the first point in time in every pixel (the first ant to enter the pixel, when there is not yet any footprint

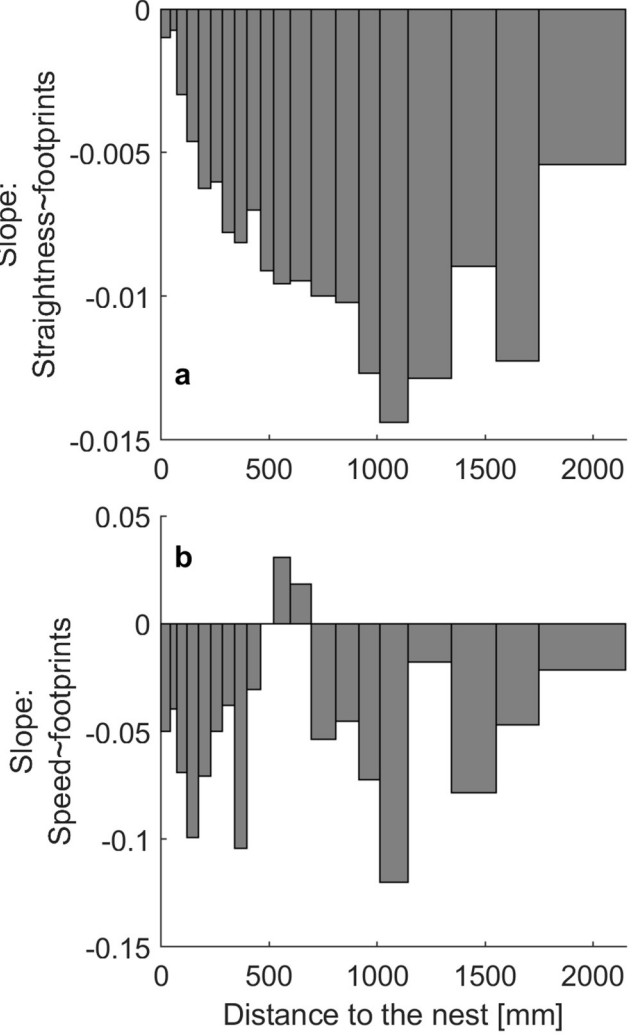

**Fig 5.** Ants walk a) straighter and b) mostly faster with lower footprint concentration, even when binning by (= controlling for) distance to the nest. Bins are sized to each contain about 9.04x10^4 points. Filled bars indicate values significantly different from 0.

information) with the final number of footprints (after 5 h) in the pixels. We find that the less straight and the slower the first ant walked over that pixel, the more footprints accumulated in the following time (Fig 6; Tables S1.6 and S1.7 in S1 File for stats; LMM of max footprint number ~ straightness of the first point, with 'nest distance' and 'colony' as random factors: Estimate = -4.6, SE = 3.56e-2, t = -129, DF = 1.78e6, p < 0.001, Lower = -4.67, Upper = -4.53; LMM of max footprint number ~ speed of the first point, with 'nest distance' and 'colony' as random factors: Estimate = 6.28e-2, SE = 2.06e-3, t = 30.44, DF = 1.8e+06, p < 0.001, Lower = 5.87e-2, Upper = 6.68e-2). This indicates that at least some of our results can be explained by ants walking less straight over some areas due to unknown factors unrelated to footprints, which leads to these areas being walked on more. Such factors might be navigational cues in the room, local changes of the surface structure, or local smells or tastes.

**Ants to move less straight on higher footprint concentrations.** The above analysis again shows causation of variation in the movement systematically changing the footprint

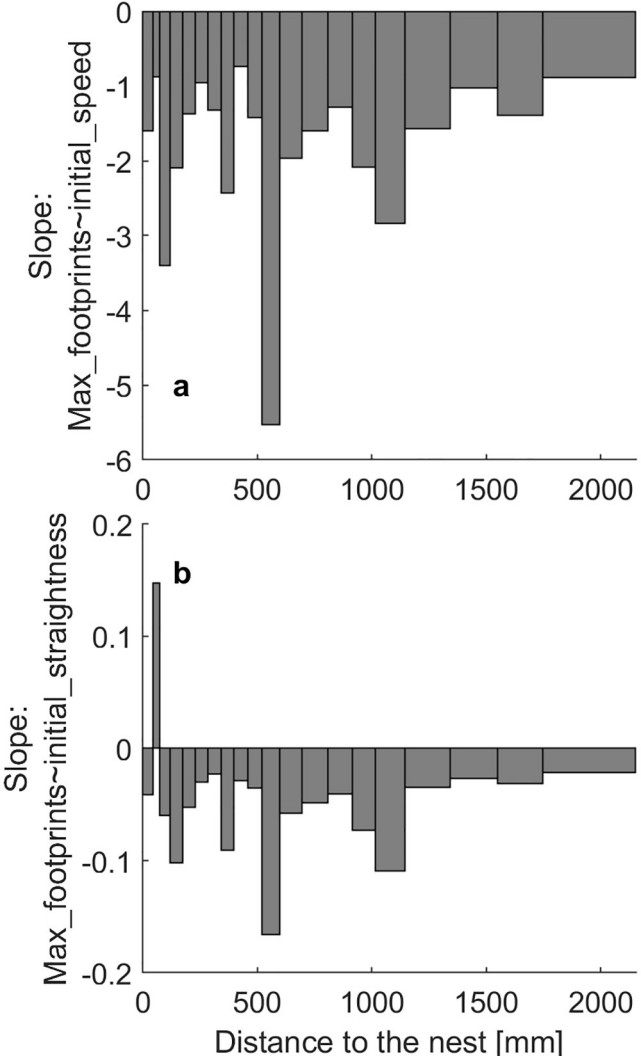

**Fig 6.** Slopes of final footprint number (after 5h) of 2x2 mm pixels over a) the straightness value and b) the speed of the first point created in that pixel (first ant entering the pixel in time), for each nest distance bin. All values are significantly different from 0.

concentration, but this does still not exclude the possibility of some effect of the footprint concentration present on the walking behavior of the ants when all other sources of variation are excluded. We thus again calculated the correlations between straightness and footprint concentration and between speed and footprint concentration, but this time while controlling for the tortuosity inducing properties of the area. We assumed the first visit to each 2x2 mm pixel to be reflective of said properties and thus binned the 2x2 mm pixels and their associated points by the straightness of the first visit to each pixel, with each bin containing the same number of points (n = 88284). We find two things: Firstly, the slopes decrease steadily, going from positive to negative values with increasing initial straightness and speed values (Fig 7; Tables S1.8 and S1.9 in S1 File for stats on individual nest distance bins). This could be explained by a 'regression to the mean' effect, where exceptionally curvy or straight tracks are likely to be followed by tracks with medium straightness, resulting in positive and negative

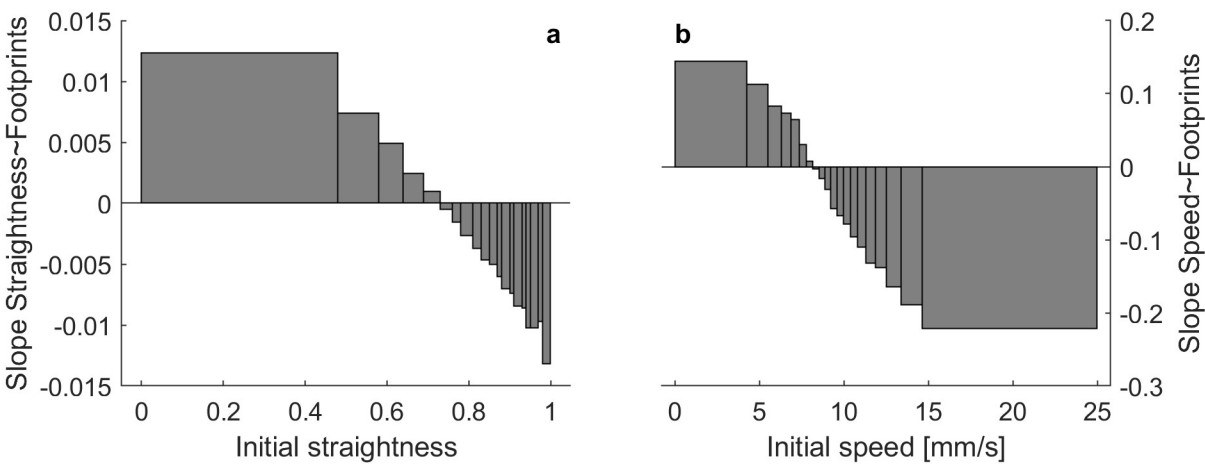

**Fig 7. Reactions to footprints, binned by 'inherent' straightness and speed of the locations.** Slope of the linear model of (straightness~footprints) over the straightness of the first point in the respective pixel. Negative values indicate ants walking straighter (or faster) on lower footprint concentrations. Bars contain the same number of points (n = 88284) and have thus different widths, as there are fewer points with lower straightness. Gray bars are significantly different from 0.

slopes, respectively. Secondly, within most nest distance bins, the slopes are negative, just like that of a LMM across all points (LMM of pooled points: straightness ~ footprints, with 'straightness of first point in that pixel', 'nest distance', and 'colony' as a random factors: Estimate = -2.46e-3, SE = 2.85e-5, t = -86.2, DF = 1.76e6, p < 0.001, Lower = -2.52–3, Upper = -2.41e-3; LMM of speed ~ footprints, with 'speed of first point', 'nest distance', and 'colony' as random factors: Estimate = -0.03, SE = 4.78e-4, t = -55.91, DF = 1.8e+06, p < 0.001, Lower = -0.028, Upper = -0.023), indicating that—all else we could think of being equal—ants will walk slightly straighter and faster off footprints. This means that in a naturalistic setting, several factors act together to create the displayed movement behavior of ants.

## Ants do not turn towards or away from pheromone

Another way ants might change their movements in reaction to footprints, which we would not have picked up in the above analysis, is by performing klinotaxis, i.e., turning into or away from a footprint gradient rather than a general change in movement pattern. This was observed in Argentine ants, who form exploratory trails through positive klinotaxis, i.e., by turning towards the side of higher footprint concentrations [40]. Negative klinotaxis could explain the more even area coverage in Hunt et al. (2020) [5], but would not necessarily change the overall track straightness. We thus replicated the analysis in Perna et al. (2012) [40]. For this, we calculated the assumed footprint concentrations separately for the front left and right quarter circles of the ant. We then used a linear model to test for a correlation between the turning direction of that point and the difference between the footprint concentration sensed by the left and right antenna. Points closer than 10 cm to the nest were excluded, as we expect ants to be in the process of exiting and entering the nest or familiarization with the environment [38] and thus ignoring footprints.

The results of the turning direction analysis show that ants did not turn preferentially in the direction of higher or lower footprint concentrations (Fig 8; Linear Model of Footprints (L)-Footprints(R) ~ turn_angle: Estimate = -3.9e-3, SE = 9.4e-3, DF = 1.78e6, t = -0.41, p = 0.68).

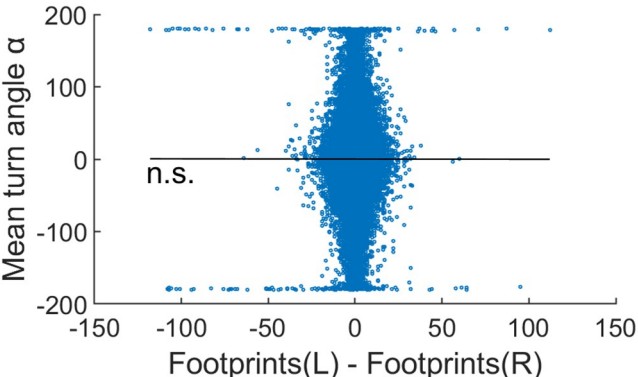

**Fig 8. Ants do not turn toward or away from a footprint gradient.** A negative slope would indicate turning toward the side of higher footprint concentrations.

## Different evaporation times do not change results qualitatively

We picked the 1 h presumed 'evaporation' time somewhat arbitrarily and wanted to ensure robustness of our results to different durations of ants still reacting to footprints. We thus repeated all analyses two more times: once while considering only those points for the footprint concentration calculation that were made less than 5 min before the focal ant crossed this spot, and once while including potentially all 5 h of the experiment, assuming no 'evaporation' effect.

We found qualitatively similar results for both of these variations with regard to straightness. For speed, we found similar results in the 5 min analyses but some minor differences when including the whole 5 h of the experiment, specifically that ants walk faster on higher footprint concentrations at medium and long distances from the nest (Fig S3.12b in S3 File), and that the points closer than 10 cm to the nest become such a large proportion that the overall positive relationship between speed and nest distance (as shown in Fig 1) flips to the negative (Fig S3.3d in S3 File). This means that overall, all findings from the 1 h analysis hold in the majority of the arena area, and only the klinokinesis effect is reversed very close and very far from the nest. All statistical analyses and figures can be found in the S2 and S3 Files.

## Replication of analysis on Hunt et al.'s data

To ensure our results are not idiosyncrasies of our setup or species, and to test whether the same effects persist when individual ants are tested without nestmates' markings present, we replicated our complete analysis methods on the openly available data used in Hunt et al. (2016; 2020) [5, 21]. While their general methods are similar to ours, there are a few key differences: they let 6 ants from each of their 3 colonies explore their arena one after another and repeated this experiment while removing the chemical footprints after each ant's trip ('cleaning', as opposed to 'no cleaning' of the other trials). They also only analyzed large-scale properties of the movement behavior (i.e., ant distributions) and walking speed, but not path straightness. We analyzed the data from the 'cleaned' trials as if there were still footprints present, to see if the pheromones are the cause of the behavior changes. Most results of their 'not cleaned' trial, i.e., with footprints present, are qualitatively the same as ours (S4 File). The only differences are that here, the slopes of the correlations between straightness and speed near the nest and the mean track distance are not significantly different from 0 (Fig S4.5e and S4.5f in S4 File), indicating that in their dataset, individual differences were either absent or the sample

size (36 tracks, compared to 1048 unique tracks in our dataset) was too small to detect them. Additionally, the correlations of speed and footprint concentration within nest distance bins do not show a clear pattern (Table S4.7 and Fig S4.12b in S4 File). Interestingly, in the data of the 'cleaning' trials there are also significant effects in the same direction as in the other here analyzed datasets, as well as a positive klinotaxis effect (S5 File).

## Discussion

### Results summary: Movement determines footprints, and footprints influence movement

We found that searching *Temnothorax rugatulus* ants walk slightly straighter and faster on lower footprint concentrations. This is opposite to the hypotheses which follow from the results of previous studies [20, 21], and opposite to what we would expect if footprints served to help ants avoid already-searched areas ('klinokinesis') in order to cover the area around their nest more evenly [5]. Straighter movement on fewer footprints, in our study, is explained by 3 factors: 1) individual differences between *T. rugatulus* ants lead to a distance selection effect, where tracks which start out straighter near the nest (and keep their straightness) moved on average farther away, where there are naturally fewer footprints. Still, even when we controlled for the distance to the nest, ants were still generally slightly less straight and slower on higher footprint concentrations. This is because of 2) spatial heterogeneity: in some areas, ants walk less straight (possibly in response to landmarks or other, local features of the environment), which leads to more footprints being created in those areas. However, even when controlling for this effect, a shallow negative correlation between straightness and footprints still persisted, indicating that 3) ants likely display klinokinesis: higher footprint concentrations lead ants to walk less straight.

A different potential mechanism explaining more even area coverage with footprints is 'klinotaxis', where ants turn into or away from higher footprint concentrations, for which we found no evidence. All our results remain qualitatively the same when changing the timescale of the presumed evaporation of pheromone footprints from 1 h to 5 min or 5 h. Our results are a clear reminder to any biologist to explicitly state the mathematical null-model assumptions of their system (e.g., which distribution is expected without the tested biological effects?). This helps to avoid overlooking non-biological correlations which alone could explain the results which otherwise would be attributed to a biological hypothesis.

### Klinokinesis: Ants walk less straight on chemical footprints

The result that ants walk slightly straighter and faster in areas with lower footprint concentration seems to imply that ants seek out already-visited areas instead of avoiding them. This is different from a previous study on *Temnothorax unifasciatus*, which showed that ants spend more time in unmarked areas and perform more U-turns towards them, although path straightness was not measured directly [20].

Below, we report on factors which, on their own, can lead to the correlation of less straight movements on more footprints. However, even when controlling for those factors, footprints still correlate with ants walking less straight. One adaptive hypothesis is that footprints indicate the locations of patchy resources. In the search for resources, straight and dispersive movements may be beneficial when searching for widely dispersed novel resources, but less straight movements allow foragers to exploit additional resources in already-discovered areas of high resource density. A switch from straight movements to or between resource patches to more tortuous movements within patches is known from different species [41] and was shown to be

mathematically efficient [42]. If ants (evolutionarily) 'expect' patchy resources, and footprints serve as cues for being in a patch, this might explain the behavior of less straight movements on more footprints. Alternatively, it could be advantageous for ants to stay in close proximity to each other when searching, for example to use the advantage of numbers when encountering a competitor, which is an important part of the ecology of this species [27]. This strategy would be similar to that of Argentine ants forming exploratory trails, which presumably let them quickly dominate any new-found resources [43, 44].

## Individual differences: Straighter ants move farther away, where there are fewer footprints

In addition to the direct effect of pheromones on ant behavior, the correlation of straighter tracks with lower footprint concentrations is also a logical result of the following 3 factors: 1) Tracks vary considerably in overall straightness. Variation in movement behavior is well documented, not only between [45], but also within colonies of ants [22, 46], and across the animal kingdom [47]. Causes for this include genetic, developmental, and experiential differences between individual ants [47–49]. 2) Straighter movements, without introducing any other biases or autocorrelations, lead to quicker displacement from the starting point [50]. 3) Footprint concentrations are expected to decrease with distance to the nest, since a) the circumference over which ants can distribute themselves increases quadratically with the distance to the nest, and b) ants are not confined to non-overlapping sectors (like pizza slices), as reported in other species [51], but rather move more like a random walk (like spaghetti). Thus, ants which are straighter to begin with move farther away from the nest, where there are fewer footprints. Most other studies on ants do not report this selection effect, possibly because they either include only a small number of ants which explore the arena individually, or over shorter time spans in smaller arenas [5, 17, 21]. We are aware of only one study which found weak evidence for a similar effect in one of 11 colonies [52]. Although we see the same straightness to footprints correlation in the Hunt et al. (2016; 2020) [5, 21] data set, here ants which move straighter near the nest do not move significantly farther away from it. However, this is likely due to the smaller sample size and thus lacking power. We argue that interindividual variation is an important feature of animal groups and may even be ecologically more important than behavioral changes of individuals in some ecological contexts. In the case at hand, ants might be searching for different kinds of targets (e.g., nest sites, food, potential nest intruders), and are thus expected to move differently according to the distributions and densities of their respective target.

## Spatial heterogeneity: Ants walk less straight in some areas, and thus deposit more pheromones there

Even when controlling for the two effects mentioned above by binning the data into pixels of 2 mm length, the small negative correlation between straightness and footprints persists. This can be explained analogously by the fact that some areas make ants walk consistently less straight than other areas. Because more tortuous walks are expected to disperse less [50], there will naturally be more footprints in these areas. Hence, a negative correlation between straightness and footprint concentration arises simply from spatial heterogeneities affecting ant movement and distribution. Some such heterogeneities in our experiment likely include subtle surface irregularities and odors as follows: The arena floor was created by covering the room floor with PVC panels taped together with adhesive tape, covered by three layers of butcher paper. Despite these layers and several months between applying the adhesive tapes and performing the experiments, ants could apparently still sense the tape below, as can be inferred

from the patterns of ant path density and straightness. As ants approached the location of the tapes, many stopped briefly or lunged backwards, and followed the edge of the (non-visible, under layers of paper) tape outline. This led to non-straight movements on areas of high footprint concentrations (Fig 1c and 1d). Another feature of the arena floor the ants apparently reacted to was a circular drain in the floor of the room, underneath the center of the arena and several additional layers of cardboard. Fewer tracks led over this circle than the adjacent areas and those that did were slightly straighter and faster. Another arena feature influencing ant movements are the walls enclosing the arena. These may have modulated ant behavior to lead them to move straighter after bumping into them and/or following them for a while. Since such very subtle features seem to have influenced the ants' behavior in our setup, it follows that in the natural habitat of the forest ground, which is orders of magnitude more patchy and heterogenous, natural odors and structural features impact the exploratory behavior of ants much more than any chemical footprints deposited by nestmates. This remarkable sensitivity to odor cues is also important to consider when interpreting lab results in the ecological context. Many lab artifacts may remain unnoticed, since we can only pick up the above described effects when pooling across all 5 colonies and hundreds of ants.

## No turning relative to footprint gradient (= no klinotaxis)

Another possibility of how ants could interact with footprint pheromones is to turn preferentially towards or away from the side of higher concentration, i.e., klinotaxis. Ants use this mechanism to follow pheromone trails [40, 53, 54], and could thus be expected to use it to increase search efficiency (by increasing collective area coverage or the evenness of coverage). However, we do not find evidence for such behavior in the analysis of our data or the data of Hunt et al. (2016; 2020) [5, 21].

## Movement to footprint correlations in previous studies

Hunt et al. (2016; 2020) [5, 21] studied the closely related, but European *Temnothorax albipennis* and reported higher speeds on footprint marked surfaces than non-marked ones. Applying our analysis methods to their openly accessible data, we found similar effects to our results within both of their treatments, except that here ants which are faster near the nest also move on average farther. Ants moving straighter on lower footprint concentration implies that although Hunt et al. (2020) [5] concluded that pheromone footprints might cause ants to cover the area around their nest more evenly on a colony-level (contrary to our conclusion), the direct effect of pheromone markings in Hunt et al. (2016; 2020) [5, 21], as in our study, is to make ants walk in a more curvy, slower pattern. Note that we cannot test directly on our data whether colonies in our experiments also explore more evenly with footprints present since we could not remove the footprints for each ant in the arena. We have picked up the effects we report in our main results possibly because we used a small-scale analysis (straightness & footprints calculated for each step), while they used large-scale analysis (mean speeds and coverage of 2 treatments). Hunt et al. (2020) [5] also do not discuss the possibility of the indirect ('non-biological') effects of expected ant distribution coupled with interindividual variation. The more even area coverage reported there and possibly present in our experiments, although not measured, might be due to a larger-scale process where ants try to avoid the general directions of high footprint concentration, based on their individual memory.

This implies that the more even ant track distribution observed in Hunt et al. (2020) [5] remains unexplained: their own individual-level data do not show either avoidance of other ants' tracks via klinotaxis (turning away) nor klinokinesis (preferentially straight movements

on pheromones leading out of densely searched areas), according to our analysis methods, the results of which support these effects.

What alternative explanations for the even distribution of tracks in Hunt et al. (2020) [5] remain? It may be that ants use individual memory of general locations of higher footprint concentrations to avoid such areas, rather than an immediate stimulus-response mechanism.

### Different movement close to the nest

Ants show generally different movement behavior than what was discussed above in the roughly 10 cm closest to the nest entrance, rapidly increasing speed and straightness with the distance to it, before reaching a plateau. This is probably mainly due to returning ants searching slowly and tortuously for the nest entrance [55, 56]. Likewise, ants leave the nest slowly and possibly gather information about the current conditions or memorize the visual scenery for orientation [38]. However, since our experiments were performed after ants had 2x5 hours to familiarize themselves, this should have a limited effect on our results. Since the ant density is relatively high near the nest, ants might also meet each other and thus stop and turn more, whether this is just to pass or to exchange information [57]. The effects of meeting ants are minimal at greater distances due to the very low rate of meeting other ants (34 interactions > 10 cm from the nest across all colonies). An important factor explaining the extremely high ant concentrations and different movement behavior close to the nest might also be that most tracks in our dataset have their mean distance to the nest at 3–7 cm (Fig 3b).

### 'Evaporation' timescale

Chemical footprints consist of multiple compounds with different evaporation times [7], making it plausible that ants react differently to footprints depending on the time since deposition. In some cases, different components of pheromone trails have been identified to have different half-life times and with it, elicit different behaviors of the followers [58]. Based on previous research [8, 40, 59] we guessed an 'evaporation' effect of 1h, which is more accurately described as the decreasing probability and intensity of reactions to footprints by ants. We did not find qualitatively different ant movement behavior when we only included footprints which were deposited less than 5 minutes prior to the ant's visit. The generally similar reactions to footprints are in line with results in Argentine ants, where behaviors did not change over 1 h of time lag [40]. In general, there is a dearth of information on the evaporation characteristics of ant pheromones, especially those making up the footprints [7]. That is mostly due to difficulties detecting the miniscule amounts of chemicals on the surfaces, or recreating naturalistic markings. Additionally, the few studies investigating this show that pheromones and their evaporation characteristics are highly variable between species, temperature, and substrate [7, 59, 60]. We expect the reactions of ants to chemical footprints to be altered with increasing time since deposition at some timescale, but as we did not find an effect of 'evaporation time' in our results, the footprint pheromones relevant to our study probably have relatively low volatility.

### Conclusion

We tested two mechanistic hypotheses about how ants react to chemical footprints. We found no evidence for klinotaxis (turning towards or away from a cline of pheromone density), but we did find evidence for klinokinesis, i.e., an effect of pheromone intensity on movement characteristics. However, this effect is inverse to what we would expect if ants avoided areas with high footprint concentrations; instead, ants seem to move in ways that make them walk more distance in such areas. We also find that the distribution of ants, and thus possibly the

distribution of collective search effort, is driven by several possibly non-intuitive effects, including interindividual variation in movement characteristics, correlations expected from random walks (i.e., straighter movements tend to lead farther away from the origin), and extreme sensitivity of ants to environmental cues, in addition to ants reacting to the footprints. Thus, habitat traversability and cues for risks and resources in the environment likely play a bigger role than the reactions to nestmate footprints in natural situations. Coordination to increase colony-level search efficiency is thus probably facilitated more through interindividual variation in movement behavior and possibly more complex individual strategies, not through pheromone avoidance. The effects we report here should be accounted for in future lab studies on how chemical footprints or home range markings alter animals' movement behavior. Additionally, behaviors of isolated individuals in well-controlled setups may not be transferable to ecologically relevant contexts and whole colonies or groups of individuals.

## Supporting information

**S1 File. Main analyses supplement.** Example track color coded by straightness values, correlation between speed and straightness per point, mean track distance to the nest, heatmaps separated by colony, statistics to Figs 5–7.
(PDF)

**S2 File. 5 min 'evaporation' time.** All main-text analyses for an assumed 5 minute 'evaporation' time.
(PDF)

**S3 File. No 'evaporation' time.** All main-text analyses for an assumed 5 h 'evaporation' time.
(PDF)

**S4 File. Hunt et al. 'NC' (footprints present).** All main-text analyses on the 'No cleaning' data of Hunt et al. 2016.
(PDF)

**S5 File. Hunt et al. 'C' (no footprints between ants).** All main-text analyses on the 'Cleaning' data of Hunt et al. 2016.
(PDF)

## Acknowledgments

We are grateful for the undergraduate students Tahsin Rasheed, Salsabeal Jarrah, Katie Perotti, and Brandy Hadley-Nihiser for their diligent manual processing of tracking data and Brian Enquist and Dan Papaj for comments on the manuscript.

## Author Contributions

**Conceptualization:** Stefan Popp, Anna Dornhaus.

**Data curation:** Stefan Popp.

**Formal analysis:** Stefan Popp.

**Funding acquisition:** Anna Dornhaus.

**Investigation:** Stefan Popp, Anna Dornhaus.

**Methodology:** Stefan Popp, Anna Dornhaus.

**Project administration:** Stefan Popp, Anna Dornhaus.

**Resources:** Anna Dornhaus.

**Software:** Stefan Popp.

**Supervision:** Anna Dornhaus.

**Validation:** Stefan Popp.

**Visualization:** Stefan Popp.

**Writing – original draft:** Stefan Popp.

**Writing – review & editing:** Stefan Popp, Anna Dornhaus.

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
