## [Decision Letter · Decision Letter 0]

19 Oct 2023

PONE-D-23-30223Searching ants walk slower and less straight on chemical footprints, but indirect effects are largePLOS ONE

Dear Dr. Popp,

Thank you for submitting your manuscript to PLOS ONE. After careful consideration, we feel that it has merit but does not fully meet PLOS ONE’s publication criteria as it currently stands. Therefore, we invite you to submit a revised version of the manuscript that addresses the points raised during the review process.

Dear authors,

I share the reviewers’ excitement and positive feedback on your manuscript. I also agree that many aspects can be clarified, expanded, or revised. I consider that all those aspects raised should be thoroughly examined. I am confident you will be able to address them with relative ease. Please pay close attention to the reviewer’s 1 mention of the wording and reasoning behind the hypothesis and the data presentation and pattern description. Also, reviewer 2 raised many important points (see, for example, comments for Line 267, Line 315, Lines 374-375 and 377.5-381, Line 386, and many others) that warrant careful consideration and likely substantial clarification or revision.

I look forward to reading your response to all comments. I consider that addressing the issues raised will yield a more compelling, concise, and clear manuscript and more smoothly highlight your data's novel contributions.

I want to add an invitation to revise the Abstract. That section would benefit from 1-2 initial sentences placing the big picture topic that your project addresses and 1-2 sentences, in the end, going back to the main implications/novelty/significance of your results. This might result in not mentioning ants in those sentences, which will allow for a closer examination of the broad implications of your work. You do a good job with this in the Introduction and Conclusion. However, that needs to be better reflected in the abstract.

We look forward to receiving your revised manuscript.

Kind regards,

Ignacio Escalante

Academic Editor

PLOS ONE

Journal Requirements:

Reviewers' comments:

Reviewer's Responses to Questions

**Comments to the Author**

1. Is the manuscript technically sound, and do the data support the conclusions?

Reviewer #1: Yes

Reviewer #2: Partly

2. Has the statistical analysis been performed appropriately and rigorously? 

Reviewer #1: Yes

Reviewer #2: Yes

3. Have the authors made all data underlying the findings in their manuscript fully available?

Reviewer #1: Yes

Reviewer #2: Yes

4. Is the manuscript presented in an intelligible fashion and written in standard English?

Reviewer #1: Yes

Reviewer #2: Yes

5. Review Comments to the Author

Reviewer #1: The authors investigate whether and how the chemical ‘footprints’ (pheromones), left behind by ants as they walk, may affect individual and collective search. They test in particular two hypotheses. First, ants may turn away from sites with higher footprint concentrations (klinotaxis), and, second, they may change their turning patterns depending on the presence of footprints (klinokinesis). They tracked 5 whole colonies of Temnothorax rugatulus ants in a large arena over 5h and approximated the footprint concentration by summing ant visitations for each point in the arena and calculated the speed and local path straightness for each point of the ant trajectories. They found that ants walk faster and straighter in areas with fewer footprints. The authors proposed two indirect explanations for this pattern: 1) ants who start out from the nest walking straighter move on average further away from the nest, where there are naturally fewer footprints, leading to an apparent relationship between footprint density and straightness, and 2) spots where ants walk less straight for another reason will accumulate more footprints, again leading to the same correlation. They interpret that the larger-scale pattern of search density around an ant colony is affected by several independent processes, including individual differences in movement pattern, local spatial heterogeneities, and ants’ reactions to chemical footprints.

I found the paper very interesting, well-designed, and easy to read. Their major strength is that the authors present several ideas that test, guiding the reader to the reasons why reject or not reject the hypotheses discussed, and showing alternative explanations in each case. Their major weakness –but not really major- is the way in which the hypotheses are written, and some figures that I found little didactic if their function is to show patterns.

Below I describe these points more in detail hoping that these suggestions help the authors to make a more didactic and clear paper.

The formulated hypotheses are in fact predictions. The real hypotheses should be de mechanisms that generate these results (i.e., turning away from sites with higher footprint concentrations, or changing their turning patterns depending on the presence of footprints). Maybe the key hypothesis is that collective search is modulated by chemical footprints in order to maximize search cover, with the associated expected results regarding turning patterns.

The title is a little confusing for me. What exactly means “…but indirect effects are large?”

What is the number of replicates? I suppose, the number of ants analyzed?

The authors refer to Fig. 1 to comment on their key result “ants moving straighter and faster when walking across fewer assumed footprints. For me, it is hard to see that pattern (i.e., a negative correlation between local straightness and footprint concentration) in the figure. The reverse x-axis does not help.

Reviewer #2: Comments for the authors:

This study by Popp and Dornhaus presents a systematic investigation of searching movement behavior in the rock crevice ant, Temnothorax rugatulus. I find this research to be of interest for its contribution to our understanding of the effects of hypothetical chemical “footprints” on forager/searching behavior in T. rugatulus, although, the lack of a “clean” arena experiment and lack of chemical evidence somewhat limits the conclusions the authors can draw. The authors present interesting results regarding the correlation between nest distance, “footprint” concentration, and ant searching behaviors, i.e., path straightness and speed, which are important factors for maximizing efficient exploration of novel space. I think this study opens questions about what chemicals (e.g., cuticular hydrocarbons, volatile compounds, or both) might be contributing to the study's findings and how these potential chemicals interact with other factors (e.g., environmental cues, intrinsic behavioral biases, and other cues/signals from nestmates, etc.). In general, the authors do a fairly good job of summarizing their findings and discussing the implications of their work and its limitations. While many of my detailed comments in the manuscript relate to suggested edits or clarifications, several comments also relate to the analyses and conclusions drawn (see questions/comments/suggestions summarized by line number below).

Questions/comments/suggestions on the main manuscript:

Line 46: Change “…in addition…” to “…additionally…”

Line 55: Change “The authors of Hunt et al. speculate…” to “The research of Hunt et al. hypothesized…”

Line 68/69: Change “…plausible that also ants might use…” to “…plausible that ants might also use…”

Line 102: Remove “…at least…”

Line 115: Change “…we filmed…” to “…we digitally video recorded…” (unless you actually video recorded them on film)

Line 116: Add commas after “ants” and “wild,” or split the information into two sentences. It’s a bit clunky to read in its current form.

Line 123: Change “…closer than ca. 5-20 cm to…” to “…less than 20 cm from…”

Line 153: Do you mean 40 mm (not 40 cm)?

Line 172: Do you mean Fig. 3a-b (not Fig. 2a-b)?

Line 177: Please indicate the specific panel of the figure for clarity, i.e., Fig. 6a.

Line 178: Consider adding “a” and “b” labels to the two panels of Fig. 7 and, again, indicate the specific panel of the figure for clarity, i.e., Fig. 7a.

Line 183: Change “…ant movements are straighter…” to “…ant movements significantly are straighter…” to indicate statistical support for the relationship.

Line 186: Do you mean Fig. S1 2a (not Fig. S1 3a)?

Lines 195-201: Please include a description of the faint vertical lines (i.e., heat maps) in panels a and c as well as the heat maps in panels b and d. I assume they are meant to represent individual data points, but these details are missing from the caption.

Line 196: Please include units for footprint concentration, e.g., units per pixel, or whatever is the accurate measurement.

Line 201: Add a comma after “i.e.” (and elsewhere throughout the manuscript).

Lines 204-205 (re: Figure 2a): Maybe you could highlight the representative bars for the two highlighted categories using either different colors or a black/grey/white combo to make them easier for the reader to identify in the results.

Lines 206-208 (re: Figure 2b): Same comment as above.

Lines 229-235: Include somewhere in the Fig. 3 caption that this is representative of all 5 colonies since not everyone will look through the supplementary figures.

Lines 233-234: Remove “The y-axis here is the x-axis in fig 1a, x-axis here is x-axis in fig 1b. This illustrates the mismatch between the x-axes in fig 1.”, as this doesn’t seem particularly necessary to mention to understand the figure.

Line 248: Change “further” to “farther”.

Line 254: Change “…(where there are fewer footprints)…” to “…(where there are also fewer footprints, Fig. 3b)…”

Line 267: The authors state, “…non-significant or positive between roughly 45 and 70 cm…”, but, according to the stats in table S1.5, bin 17 (roughly 1200-1300 mm) is also not significant.

Line 272: The caption states, “(only one bar is unfilled…”, but, again, bin 17 on Figure 5b should be unfilled as well.

Line 285: Please indicate the specific panel of the figure for clarity, i.e., Fig. 6a.

Lines 291-293: Since the speed vs. footprint relationship was not statistically relevant, Fig. 6b might be better in the supplementary materials.

Line 301: Change “Higher footprint concentrations cause ants to move less straight” to “Ants move less straight on higher footprint concentrations”. Claiming causality based on correlative results (even if there is a causal relationship) is inappropriate given the lack of experimental treatments in this study.

Line 309: The authors state, “…with bin widths of 0.1 straightness…”, but the bin widths appear to vary from less than 0.1 up to 0.5. Is this a typo or am I misinterpreting this? Also, the caption for Figure 7 reads, "...bars contain the same number of points (n = 88931) and are thus differently sized..." (Lines 320-321) so there seems to be contradictory info.

Line 310: Please indicate the specific panel of the figure for clarity, i.e., Fig. 7a.

Line 315: The summary of Figure 7b is completely missing here. While the results at lower initial speeds (<10 mm/s) largely match predictions, the results at the initial higher speeds (ca. 15-25 mm/s) have a significantly positive correlation between speed and footprints). This seems especially unusual given the prediction that more footprints lead to slower-moving ants and vice versa. What do you think is going on here? Is there a way to determine where these points cluster in space? Perhaps some uncontrolled factor is at play?

Line 339: This sentence reads a little abruptly. Consider adding something like, "The results of the turning direction analysis show that..." to the beginning of the statement.

Line 355: Change “short” to “intermediate”.

Line 355: Do you mean Fig. S3.12b (not Fig. S3.5)?

Line 358: Do you mean excluded (instead of included)?

Line 358: Please indicate the specific panel of the figure for clarity, i.e., Fig. S3.3d.

Line 364-365: Change “…to test whether our effects persist…” to “…to test whether the same effects persist…” since these are not data from your study.

Line 366: Add “methods” after analysis.

Lines 366-368: For clarity, please include a brief summary of the similarities/differences of the Hunt et al. studies/experiments to your study.

Line 370: Please indicate the specific panels of the figure for clarity, i.e., Fig. S4.5e & f.

Line 372: Do you mean Fig. S4.12b (not Fig. S4.4b)?

Lines 374-375 and 377.5-381: Most of this is an interpretation of the results, which should go into the discussion or be removed if it is already present in the discussion.

Lines 376-377.5: This can move to the beginning of the paragraph to help explain the context, i.e., see earlier comment for lines 366-368.

Line 386: It seems that the authors of reference 20 didn’t directly investigate the relationship between straightness/speed and footprint concentration, so this claim should be reworded to reflect the differences/similarities more accurately between your findings and this study.

Line 389: Add “T. rugatulus” before “ants” for extra clarity.

Line 397: Change “…ants display klinokinesis…” to “…ants likely display klinokinesis…” since the conclusion is based on correlative results.

Line 400: Remove “also”.

Line 403: Change “warning” to “reminder”.

Line 404: Add a comma after “e.g.”.

Line 408: Change “Chemical footprints make ants walk less straight” to “Ants walk less straight on chemical footprints”. See earlier comment for line 301.

Line 411: Change “…previous study on another species in the same genus…” to “…a previous study on Temnothorax albipennis,...”.

Lines 411-412: The authors state, “…ants spend more time in unmarked areas”, but (unless I’m mistaken) the amount of time spent in each area versus footprint concentration was not measured in this study or your study, so is it correct to make this claim/comparison? In theory, couldn't ants walk faster and straighter on lower footprint concentrations and spend more total time in lower footprint concentrations?

Line 416: Change”… seem to make ants walk less straight.” to “…correlate with ants walking less straight.”

Line 428: Change “instantaneously” to “quickly”.

Line 431: Change “pheromone” to “pheromones”

Line 451: Add a comma after “e.g.”.

Line 454: Change “Spatial heterogeneity: Some areas make ants walk less straight…” to “Spatial heterogeneity: Ants walk less straight in some areas…”.

Line 486: Change “which is also called” to “i.e.,”.

Line 489: Change “…we do not find such behavior in our data…” to “…we do not find evidence for such behavior in the analysis of our data…”.

Line 494: Add “methods” after analysis.

Line 494: This second part of this sentence is difficult to follow, i.e., “…we found similar effects to our results within trials, except that here ants…”. Do you mean “we found similar effects in our results for within treatment analyses (i.e., "cleaning" [C] and "no cleaning" [NC] treatments), except that in our study ants…”? If not, please clarify.

Line 507: Change “…our experiments might be due to…” to “…our experiments, although not measured, might be due to…”.

Line 508: Change “private” to “individual”.

Line 512: Change “…according to our analysis (and our data support of these effects).” to “…according to our analysis methods, the results of which support of these effects.”.

Line 525: Do you mean 2 x 24 hours (not 2x5 hours)? The methods mentioned that day 3 recordings were used.

Lines 553-555: Change “We found no evidence for klinotaxis (turning towards or away from a cline of pheromone density). We found klinokinesis, i.e. an…” to “We found no evidence for klinotaxis (turning towards or away from a cline of pheromone density), but we did find evidence for klinokinesis, i.e., an…”

Lines 557-558: The authors state, “…ants seem to move in ways that make them spend more time in such areas.”, which is speculative, but the claim would have more power if the study also included an analysis of the amount of time spent in an area vs. footprint concentration.

Line 560: Please be more specific about what is meant by “geometric correlations”.

Questions/comments/suggestions on the supplemental materials:

Figure S3.3 caption: The authors state, “The majority of ants walk e) straighter and f) faster on lower footprint concentrations…”, but this implies that more than 50% of tracks were straighter and or faster on lower "footprint concentrations". The results seem to show that more than 50% showed no significant correlation between straightness and "footprint concentration" and more than 78% showed no significant correlation between speed and "footprint concentration." Please reword this to more accurately reflect the observed pattern.

Figure S3.5: Shift panels c-f down so that the heatmap legend values don't overlap with the two panels above.

Figure S3.12a: This figure panel is missing the numerical values and tic-marks on the x-axis.

Table S4.4 caption: Do you mean “faster” (instead of slower)? Also, note that this statement is based on a statistically insignificant result.

Table S4.10 caption: This claim is only true for the results of the first bin. Consider rewording to reflect the findings more accurately.

Table S4.11 caption: This caption should refer to speed vs. footprints and note that the relationship is more complicated.

Figure S5.1: The figure label/number should be S5.3 (according to references in Tables S5.1 & S5.2 above). Also, as in Figure S3.3, the authors state, “The majority of ants walk e) straighter and f) faster on lower footprint concentrations…”, but, again, this implies that more than 50% of tracks were straighter and or faster on lower "footprint concentrations". The results seem to show that 50% showed no significant correlation between straightness and "footprint concentration" and more than 58% showed no significant correlation between speed and "footprint concentration." Please reword this to reflect the observed pattern more accurately.

Table S5.4 caption: This claim is not statistically significant according to the results. Consider rewording to reflect the findings more accurately.

Figure S5.5: Shift panels c-f down so that the heatmap legend values don't overlap with the two panels above.

6. PLOS authors have the option to publish the peer review history of their article (what does this mean?). If published, this will include your full peer review and any attached files.

Reviewer #1: **Yes: **Alejandro G. Farji-Brener

Reviewer #2: No

---

## [Author Response · Author response to Decision Letter 0]

26 Dec 2023

Dear Editor,

Thank you for your positive evaluation of our work and we are happy to address your and the

reviewer’s comments.

Below are in blue our responses to the comments.

Unprompted changes:

We double-checked the cause of the small peak around 800 cm in Fig. 3b and removed 313

points which were false positive tracking errors and had not been removed before. Hence, we

reran all analyses which contain these data points.

Fig. 6: We show in Fig. 5 that one should control for nest distance effects but do not take this

into account in (the results of) Fig. 6. We thus changed Fig. 6 from line plots showing the

correlations regardless of nest distance, to bar plots with correlations of points separated into

nest distance bins, analogous to Fig. 5.

Heatmaps: We changed the appearance of the pixel in all heatmaps, since they had been

displayed too big and thus overlapping.

Editor

I want to add an invitation to revise the Abstract. That section would benefit from 1-2 initial

sentences placing the big picture topic that your project addresses and 1-2 sentences, in the

end, going back to the main implications/novelty/significance of your results. This might

result in not mentioning ants in those sentences, which will allow for a closer examination

of the broad implications of your work. You do a good job with this in the Introduction and

Conclusion. However, that needs to be better reflected in the abstract.

We changed the first few sentences of the abstract and added an implications sentence at the

end.

Reviewer #1:

I found the paper very interesting, well-designed, and easy to read. Their major strength is

that the authors present several ideas that test, guiding the reader to the reasons why reject

or not reject the hypotheses discussed, and showing alternative explanations in each case.

Their major weakness –but not really major- is the way in which the hypotheses are written,

and some figures that I found little didactic if their function is to show patterns.

Below I describe these points more in detail hoping that these suggestions help the authors

to make a more didactic and clear paper.

We thank you for your helpful comments and believe that our changes to the title and figure 1

made this a clearer manuscript.

The formulated hypotheses are in fact predictions. The real hypotheses should be the

mechanisms that generate these results (i.e., turning away from sites with higher footprint

concentrations, or changing their turning patterns depending on the presence of footprints).

Maybe the key hypothesis is that collective search is modulated by chemical footprints in

order to maximize search cover, with the associated expected results regarding turning

patterns.

We agree that the collective search modulation is the overarching (ultimate) hypothesis, but

view klinotaxis and klinokinesis as two mechanistic sub-hypotheses, from which the predictions

of turn angle bias and different straightnesses follow.

The title is a little confusing for me. What exactly means “…but indirect effects are large?”

For maximum clarity, we changed the title to “Searching ants do not avoid chemical footprints,

but geometric artifacts have large effects and make causal effects of pheromone hard to detect

What is the number of replicates? I suppose, the number of ants analyzed?

We added a sentence in the General setup section with these numbers.

The authors refer to Fig. 1 to comment on their key result “ants moving straighter and faster

when walking across fewer assumed footprints. For me, it is hard to see that pattern (i.e., a

negative correlation between local straightness and footprint concentration) in the figure.

The reverse x-axis does not help.

We zoomed in on the y-axis to better show the slope without cutting off too much important

information. We chose to keep the x-axis reversed for better compatibility between the two sides

of this figure.

Reviewer #2:

This study by Popp and Dornhaus presents a systematic investigation of searching

movement behavior in the rock crevice ant, Temnothorax rugatulus. I find this research to be

of interest for its contribution to our understanding of the effects of hypothetical chemical

“footprints” on forager/searching behavior in T. rugatulus, although, the lack of a “clean”

arena experiment and lack of chemical evidence somewhat limits the conclusions the

authors can draw. The authors present interesting results regarding the correlation between

nest distance, “footprint” concentration, and ant searching behaviors, i.e., path straightness

and speed, which are important factors for maximizing efficient exploration of novel space. I

think this study opens questions about what chemicals (e.g., cuticular hydrocarbons,

volatile compounds, or both) might be contributing to the study's findings and how these

potential chemicals interact with other factors (e.g., environmental cues, intrinsic behavioral

biases, and other cues/signals from nestmates, etc.). In general, the authors do a fairly

good job of summarizing their findings and discussing the implications of their work and its

limitations. While many of my detailed comments in the manuscript relate to suggested

edits or clarifications, several comments also relate to the analyses and conclusions drawn

(see questions/comments/suggestions summarized by line number below).

We thank you for your extremely thorough and helpful review. Your comments made us aware

of an error in one of our analyses and they helped us describe our study more clearly and

accurately. We traded off a ‘clean’ arena treatment with the large number of ants simultaneously

tracked, as it would have been unfeasible to clean the arena after each exploration bout for

hundreds or thousands of exploration bouts. Our approach lead us to detect the effects of

interindividual variation and sensitivity to environmental cues, which we regard as perhaps the

most important outcomes of this study.

Questions/comments/suggestions on the main manuscript:

Line 46: Change “…in addition…” to “…additionally…”

Done.

Line 55: Change “The authors of Hunt et al. speculate…” to “The research of Hunt et al.

hypothesized…”

Done.

Line 68/69: Change “…plausible that also ants might use…” to “…plausible that ants might

also use…”

Done also.

Line 102: Remove “…at least…”

Done.

Line 115: Change “…we filmed…” to “…we digitally video recorded…” (unless you actually

video recorded them on film)

Since a) the commonly understood meaning of ‘to film’ has expanded from analogue to digital

video recordings with the advent of the latter, and b) “digitally video recorded” unnecessarily

complicates the sentence, we would like to stick with our initial phrasing.

Line 116: Add commas after “ants” and “wild,” or split the information into two sentences.

It’s a bit clunky to read in its current form.

Split into two sentences.

Line 123: Change “…closer than ca. 5-20 cm to…” to “…less than 20 cm from…”

Done.

Line 153: Do you mean 40 mm (not 40 cm)?

Yes! Done.

Line 172: Do you mean Fig. 3a-b (not Fig. 2a-b)?

Done.

Line 177: Please indicate the specific panel of the figure for clarity, i.e., Fig. 6a.

Done.

Line 178: Consider adding “a” and “b” labels to the two panels of Fig. 7 and, again, indicate

the specific panel of the figure for clarity, i.e., Fig. 7a.

Done.

Line 183: Change “…ant movements are straighter…” to “…ant movements significantly are

straighter…” to indicate statistical support for the relationship.

Done.

Line 186: Do you mean Fig. S1 2a (not Fig. S1 3a)?

Yes! Done.

Lines 195-201: Please include a description of the faint vertical lines (i.e., heat maps) in

panels a and c as well as the heat maps in panels b and d. I assume they are meant to

represent individual data points, but these details are missing from the caption.

Done: These are both kernel density estimations of raw points.

Line 196: Please include units for footprint concentration, e.g., units per pixel, or whatever is

the accurate measurement.

Done.

Line 201: Add a comma after “i.e.” (and elsewhere throughout the manuscript).

Done.

Lines 204-205 (re: Figure 2a): Maybe you could highlight the representative bars for the two

highlighted categories using either different colors or a black/grey/white combo to make

them easier for the reader to identify in the results.

Done:

Lines 206-208 (re: Figure 2b): Same comment as above.

Done.

Lines 229-235: Include somewhere in the Fig. 3 caption that this is representative of all 5

colonies since not everyone will look through the supplementary figures.

Done.

Lines 233-234: Remove “The y-axis here is the x-axis in fig 1a, x-axis here is x-axis in fig 1b.

This illustrates the mismatch between the x-axes in fig 1.”, as this doesn’t seem particularly

necessary to mention to understand the figure.

Done.

Line 248: Change “further” to “farther”.

Furthermore done.

Line 254: Change “…(where there are fewer footprints)…” to “…(where there are also fewer

footprints, Fig. 3b)…”

Also done.

Line 267: The authors state, “…non-significant or positive between roughly 45 and 70 cm…”,

but, according to the stats in table S1.5, bin 17 (roughly 1200-1300 mm) is also not

significant.

Added.

Line 272: The caption states, “(only one bar is unfilled…”, but, again, bin 17 on Figure 5b

should be unfilled as well.

Done.

Line 285: Please indicate the specific panel of the figure for clarity, i.e., Fig. 6a.

Done.

Lines 291-293: Since the speed vs. footprint relationship was not statistically relevant, Fig.

6b might be better in the supplementary materials.

We think that it benefits the reader by keeping all figures in the same format (i.e., straightness

left and speed right), making it easier to identify the patterns.

Line 301: Change “Higher footprint concentrations cause ants to move less straight” to

“Ants move less straight on higher footprint concentrations”. Claiming causality based on

correlative results (even if there is a causal relationship) is inappropriate given the lack of

experimental treatments in this study.

Done.

Line 309: The authors state, “…with bin widths of 0.1 straightness…”, but the bin widths

appear to vary from less than 0.1 up to 0.5. Is this a typo or am I misinterpreting this? Also,

the caption for Figure 7 reads, "...bars contain the same number of points (n = 88931) and

are thus differently sized..." (Lines 320-321) so there seems to be contradictory info.

We changed that Results sentence to reflect that, in fact, bins contain the same number of

points. We clarified that the bin width in the figures are different, while containing the same

number of points.

Line 310: Please indicate the specific panel of the figure for clarity, i.e., Fig. 7a.

Done.

Line 315: The summary of Figure 7b is completely missing here. While the results at lower

initial speeds (<10 mm/s) largely match predictions, the results at the initial higher speeds

(ca. 15-25 mm/s) have a significantly positive correlation between speed and footprints).

This seems especially unusual given the prediction that more footprints lead to

slower-moving ants and vice versa. What do you think is going on here? Is there a way to

determine where these points cluster in space? Perhaps some uncontrolled factor is at

play?

Extra thanks for bringing this up: When looking into this, we spotted a bug which resulted in the

pixels for the analyses of Fig. 6 and 7 being much smaller than 2 mm. We thus updated those

two figures (more on Fig. 6 at the end of this document) and the results text. In short, we see a

pattern of steadily decreasing slopes with increasing initial straightness and speed, which can

be explained with a ‘regression to the mean’ effect.

Line 339: This sentence reads a little abruptly. Consider adding something like, "The results

of the turning direction analysis show that..." to the beginning of the statement.

Done.

Line 355: Change “short” to “intermediate”.

Done

Line 355: Do you mean Fig. S3.12b (not Fig. S3.5)?

Yes! Done.

Line 358: Do you mean excluded (instead of included)?

No, we mean to say that the correlation flips if nothing is excluded, and removed the

unnecessary last part of the sentence.

Line 358: Please indicate the specific panel of the figure for clarity, i.e., Fig. S3.3d.

Done.

Line 364-365: Change “…to test whether our effects persist…” to “…to test whether the same

effects persist…” since these are not data from your study.

Done.

Line 366: Add “methods” after analysis.

Done.

Lines 366-368: For clarity, please include a brief summary of the similarities/differences of

the Hunt et al. studies/experiments to your study.

We added/changed this section: “While their general methods are similar to ours, there are a

few key differences: they let 6 ants from each of their 3 colonies explore their arena one

after another, and repeated this experiment while removing the chemical footprints after

each ant’s trip (‘cleaning’, as opposed to ‘no cleaning’ in the other trials). They also only

analyzed large-scale properties of the movement behavior (i.e., ant distributions) and

walking speed, but not path straightness.”

Line 370: Please indicate the specific panels of the figure for clarity, i.e., Fig. S4.5e & f.

Done.

Line 372: Do you mean Fig. S4.12b (not Fig. S4.4b)?

Yes! Done.

Lines 374-375 and 377.5-381: Most of this is an interpretation of the results, which should

go into the discussion or be removed if it is already present in the discussion.

We changed the first sentence in accordance with the changes in response to your next

comment and removed the second sentence.

Lines 376-377.5: This can move to the beginning of the paragraph to help explain the

context, i.e., see earlier comment for lines 366-368.

Done.

Line 386: It seems that the authors of reference 20 didn’t directly investigate the relationship

between straightness/speed and footprint concentration, so this claim should be reworded

to reflect the differences/similarities more accurately between your findings and this study.

We changed the sentence to reflect that the results of the cited studies lead to these

hypotheses.

Line 389: Add “T. rugatulus” before “ants” for extra clarity.

Done.

Line 397: Change “…ants display klinokinesis…” to “…ants likely display klinokinesis…” since

the conclusion is based on correlative results.

Done.

Line 400: Remove “also”.

Done.

Line 403: Change “warning” to “reminder”.

Done.

Line 404: Add a comma after “e.g.”.

Done.

Line 408: Change “Chemical footprints make ants walk less straight” to “Ants walk less

straight on chemical footprints”. See earlier comment for line 301.

Done.

Line 411: Change “…previous study on another species in the same genus…” to “…a previous

study on Temnothorax albipennis,...”.

Done.

Lines 411-412: The authors state, “…ants spend more time in unmarked areas”, but (unless

I’m mistaken) the amount of time spent in each area versus footprint concentration was not

measured in this study or your study, so is it correct to make this claim/comparison? In

theory, couldn't ants walk faster and straighter on lower footprint concentrations and spend

more total time in lower footprint concentrations?

This reference points to a study on Temnothorax unifasciatus, not one of the Hunt et al. papers.

In that study, the authors analyzed time spent in different areas.

Line 416: Change”… seem to make ants walk less straight.” to “…correlate with ants walking

less straight.”

Done.

Line 428: Change “instantaneously” to “quickly”.

Quickly done.

Line 431: Change “pheromone” to “pheromones”

Done.

Line 451: Add a comma after “e.g.”.

Done.

Line 454: Change “Spatial heterogeneity: Some areas make ants walk less straight…” to

“Spatial heterogeneity: Ants walk less straight in some areas…”.

Done.

Line 486: Change “which is also called” to “i.e.,”.

Done.

Line 489: Change “…we do not find such behavior in our data…” to “…we do not find evidence

for such behavior in the analysis of our data…”.

Done.

Line 494: Add “methods” after analysis.

Done.

Line 494: This second part of this sentence is difficult to follow, i.e., “…we found similar

effects to our results within trials, except that here ants…”. Do you mean “we found similar

effects in our results for within treatment analyses (i.e., "cleaning" [C] and "no cleaning" [NC]

treatments), except that in our study ants…”? If not, please clarify.

We changed “within trials” to “in both of their treatments”.

Line 507: Change “…our experiments might be due to…” to “…our experiments, although not

measured, might be due to…”.

Done.

Line 508: Change “private” to “individual”.

Done.

Line 512: Change “…according to our analysis (and our data support of these effects).” to

“…according to our analysis methods, the results of which support of these effects.”.

Done.

Line 525: Do you mean 2 x 24 hours (not 2x5 hours)? The methods mentioned that day 3

recordings were used.

The ants were kept inside their nests (i.e. not allowed into the arena) for the time between the

5-h-long trials. They thus only had 2x5 h to familiarize themselves.

Lines 553-555: Change “We found no evidence for klinotaxis (turning towards or away from

a cline of pheromone density). We found klinokinesis, i.e. an…” to “We found no evidence for

klinotaxis (turning towards or away from a cline of pheromone density), but we did find

evidence for klinokinesis, i.e., an…”

Done.

Lines 557-558: The authors state, “…ants seem to move in ways that make them spend

more time in such areas.”, which is speculative, but the claim would have more power if the

study also included an analysis of the amount of time spent in an area vs. footprint

concentration.

We changed the phrasing to “walk more distance”, since we are less interested in the time

spent, than the area explored.

Line 560: Please be more specific about what is meant by “geometric correlations”.

We changed this to say “correlations expected from random walks (i.e., straighter movements

tend to lead farther away from the origin” to indicate biologically non-interesting correlations.

Questions/comments/suggestions on the supplemental materials:

Figure S3.3 caption: The authors state, “The majority of ants walk e) straighter and f) faster

on lower footprint concentrations…”, but this implies that more than 50% of tracks were

straighter and or faster on lower "footprint concentrations". The results seem to show that

more than 50% showed no significant correlation between straightness and "footprint

concentration" and more than 78% showed no significant correlation between speed and

"footprint concentration." Please reword this to more accurately reflect the observed

pattern.

Fixed.

Figure S3.5: Shift panels c-f down so that the heatmap legend values don't overlap with the

two panels above.

Fixed.

Figure S3.12a: This figure panel is missing the numerical values and tick-marks on the

x-axis.

Fixed.

Table S4.4 caption: Do you mean “faster” (instead of slower)? Also, note that this statement

is based on a statistically insignificant result.

Fixed, reporting no significant effects.

Table S4.10 caption: This claim is only true for the results of the first bin. Consider

rewording to reflect the findings more accurately.

Fixed, after rerunning the analysis.

Table S4.11 caption: This caption should refer to speed vs. footprints and note that the

relationship is more complicated.

Fixed, after rerunning the analysis.

Figure S5.1: The figure label/number should be S5.3 (according to references in Tables S5.1

& S5.2 above). Also, as in Figure S3.3, the authors state, “The majority of ants walk e)

straighter and f) faster on lower footprint concentrations…”, but, again, this implies that

more than 50% of tracks were straighter and or faster on lower "footprint concentrations".

The results seem to show that 50% showed no significant correlation between straightness

and "footprint concentration" and more than 58% showed no significant correlation between

speed and "footprint concentration." Please reword this to reflect the observed pattern more

accurately.

Fixed.

Table S5.4 caption: This claim is not statistically significant according to the results.

Consider rewording to reflect the findings more accurately.

Fixed.

Figure S5.5: Shift panels c-f down so that the heatmap legend values don't overlap with the

two panels above.

Fixed.

---

## [Decision Letter · Decision Letter 1]

25 Jan 2024

PONE-D-23-30223R1Searching ants do not avoid chemical footprints, but geometric artifacts have large effects and make causal effects of pheromone hard to detectPLOS ONE

Dear Dr. Popp,

Thank you for submitting your manuscript to PLOS ONE. After careful consideration, we feel that it has merit but does not fully meet PLOS ONE’s publication criteria as it currently stands. Therefore, we invite you to submit a revised version of the manuscript that addresses the points raised during the review process.

We look forward to receiving your revised manuscript.

Kind regards,

Liezl Ularte Callo

Support Staff - Editorial

PLOS ONE

Reviewers' comments:

Reviewer's Responses to Questions

**Comments to the Author**

1. If the authors have adequately addressed your comments raised in a previous round of review and you feel that this manuscript is now acceptable for publication, you may indicate that here to bypass the “Comments to the Author” section, enter your conflict of interest statement in the “Confidential to Editor” section, and submit your "Accept" recommendation.

Reviewer #1: All comments have been addressed

Reviewer #3: (No Response)

2. Is the manuscript technically sound, and do the data support the conclusions?

Reviewer #1: Yes

Reviewer #3: Partly

3. Has the statistical analysis been performed appropriately and rigorously? 

Reviewer #1: N/A

Reviewer #3: I Don't Know

4. Have the authors made all data underlying the findings in their manuscript fully available?

Reviewer #1: Yes

Reviewer #3: No

5. Is the manuscript presented in an intelligible fashion and written in standard English?

Reviewer #1: Yes

Reviewer #3: Yes

6. Review Comments to the Author

Reviewer #1: I just read the second version of the manuscript now entitled “Searching ants do not avoid chemical footprints, but geometric artifacts have large effects and make causal effects of pheromone hard to detect”.

This version seems to me better than the previous one. The authors answered all my concerns successfully and, thus, I do not have major comments. My only observation is that I still found the title a little confusing. I recommend rewriting it to emphasize the take-home message of the work simply and more directly. Maybe a version of the first subtitle of the discussion section such as “Movement determines footprints, and footprints influence movement: collective searching in the ant…” (or something like that) may work

Reviewer #3: In this work, St. Popp and Dornhaus examine the role of previous ant presence on ant movement decisions. Specifically, they reasonably assume that ants passively deposit cuticular hydrocarbons (CHC) as they walk, and after allowing Temnothorax ants to walk in a large arena for 5 hours, correlated speed and path straightness with (presumed) CHC concentrations. They report that ants walk (slightly) faster and straighter on lower CHC levels. They highlight several non-biological explanations for this, but report that some effect remains even when controlling for the non-biological effects.

This is the first time I am reviewing this manuscript. I note that this is a resubmission, but have not looked at the previous reviewer comments or responses, to remain unbiased by them. I am not qualified to review the technical aspects of this work (tracking with Trex, data processing). As this manuscript was submitted to PLoS One, I will refrain from commenting on the subjective interest level of this work.

The work asks reasonable questions, re-examining a topic others have looked at before, but with an unusually rich dataset, and with a more critical eye to non-biological explanations. The writing is generally clear. Some of the methods require a bit more explanation (See detailed comments). The conclusions broadly follow from the results.

I have two major concerns – one technical, and one conceptual.

The technical concern concerns the procedure used to ‘control’ for innate attractiveness of a patch, and thus for the non-biological explanations. The authors “assumed the first visit to each 2x2 mm pixel to be reflective of said properties” (line 318). I don't think this is a robust enough assumption to hang a predictive claim on. I'm not saying there is a better option - I can't think of one. But even if there is a correlation between the first pixel visit and some property of the pixel, it will be so noisy as to be almost not there - purely descriptive. I don’t think it’s reasonable to write unqualified claims about this in the abstract of the manuscript.

My second, conceptual, issue, regards effect sizes. These are indeed reported, but not on any useful, easily-comprehensible scale. I strongly advise providing comprehensible effect size descriptions, such as “the top 10% most visited pixels had XXX% slower movement speeds and XXX% lower sinuosity than the average pixel” or something similar. I will provide some further examples in the detailed comments. Looking at the figures, it seems that the effect sizes are uniformly very small. This (if it is the case) needs to be highlighted in the text, especially the abstract and discussion, by the addition of adjectives such as “slightly”, “small”, and “minor”. This strongly changes what readers will take away from the manuscript. In the conclusions, the authors make some very important statements, which I think are otherwise hidden: that various non-interactive effects (such as environmental idiosyncrasies) are likely to have an overwhelmingly larger effect of movement patterns than any putative ‘home range marking pheromone’ effect. I think this experiment shows that very nicely! But this barely comes out in the abstract. Rather, the abstract focusses on, to me, possibly biologically unimportant effects, while ignoring the more interesting message. All if this is hard to be certain of without some clearer description of the effect sizes.

I also found the discussion, and indeed the whole paper, a bit on the long and rambling side. However, this is a personal taste thing.

Overall, the study is well designed and well suited to filling a gap in the literature. I have no doubt that it should be published, but am a bit concerned that the results do not strongly support all the conclusions drawn.

MINOR COMMENTS

Title – “geometric artifacts” is not intuitive.

Line 81 – I think a lot of readers might be confused when calling these CHCs pheromones. It’s a reasonable term, but readers are used to thinking of trail or alarm pheromones in ants. Perhaps be explicit first that you are terming them ‘pheromones’. Also somewhere it is important to explicitly mention that this species does not use a recruitment pheromone.

Line 127 5-20cm is a big range! Why such a range?

Lines 130-132 – This was not easy for me to understand. Moreover, it was not clear why this was done. This seems like a major alteration to the data, so the motivation should be clear.

Line 153 – reanalysis with 5 minutes and 5 hours – excellent! A really robust approach. I applaud this.

Line 167 – was ‘ant’ included as a random effect, since ants provide multiple paths?

Line 170 – Matlab, eh? Can this produce a produce analysis document? Like an knitted Rmarkdown file? Would be good in the spirit of open science. I note that as the data was also provided as part of the matlab file, I cannot access it or examine it, since I do not own Matlab. R is free and open source, you know?

Table 1 and elsewhere – “st~FP” is meaningless. Please use intelligible terms. I assume FP is footprints? St is straightness?. Same for lines 250 and elsewhere

Line 201 – only a monster would separate the figure legends from the figures, and have the figures hidden in the back. You’re not a monster, are you? I can only assume PLoS One insisted on this. They, then, must be the monsters. No, but seriously, please next time keep the figures with the legends in the main text.

Lines 212-217 – this is an excellent paragraph; useful and clear. Can you please provide these sort of summary statistics for the collective effects?

Line 251 – here for example adding something like “ants moved XXX% faster away from the nest”. But throughout please illustrate the effect sizes.

Lines 294 – is this because most pixels only ever get walked over once? Maybe we are seeing “slower more torturous ants spend longer at a location"?

Line 311 – delete ‘to’

Line 392 – some sort of typo

Lines 405 “…ants walk a tiny bit straighter and faster…” etc

Lines 496-500 – I agree completely, and feel like this should be a bigger message of the paper.

Lines 518 and thereafter – some citation issues going on here (2020)(5).

Line 575 - small b

7. PLOS authors have the option to publish the peer review history of their article (what does this mean?). If published, this will include your full peer review and any attached files.

Reviewer #1: No

Reviewer #3: **Yes: **Tomer J. Czaczkes

---

## [Author Response · Author response to Decision Letter 1]

7 Feb 2024

See uploaded document for proper formatting.

Our responses are in blue.

Reviewer #1: I just read the second version of the manuscript now entitled “Searching ants do

not avoid chemical footprints, but geometric artifacts have large effects and make causal

effects of pheromone hard to detect”.

This version seems to me better than the previous one. The authors answered all my concerns

successfully and, thus, I do not have major comments. My only observation is that I still found

the title a little confusing. I recommend rewriting it to emphasize the take-home message of the

work simply and more directly. Maybe a version of the first subtitle of the discussion section

such as “Movement determines footprints, and footprints influence movement: collective

searching in the ant…” (or something like that) may work

Thank you for your positive evaluation and the suggestion for a great title, which we adapted

slightly modified.

Reviewer #3: In this work, St. Popp and Dornhaus examine the role of previous ant presence on

ant movement decisions. Specifically, they reasonably assume that ants passively deposit

cuticular hydrocarbons (CHC) as they walk, and after allowing Temnothorax ants to walk in a

large arena for 5 hours, correlated speed and path straightness with (presumed) CHC

concentrations. They report that ants walk (slightly) faster and straighter on lower CHC levels.

They highlight several non-biological explanations for this, but report that some effect remains

even when controlling for the non-biological effects.

This is the first time I am reviewing this manuscript. I note that this is a resubmission, but have

not looked at the previous reviewer comments or responses, to remain unbiased by them. I am

not qualified to review the technical aspects of this work (tracking with Trex, data processing).

As this manuscript was submitted to PLoS One, I will refrain from commenting on the subjective

interest level of this work.

The work asks reasonable questions, re-examining a topic others have looked at before, but with

an unusually rich dataset, and with a more critical eye to non-biological explanations. The

writing is generally clear. Some of the methods require a bit more explanation (See detailed

comments). The conclusions broadly follow from the results.

We are grateful for your helpful and insightful comments and believe that addressing these has

increased the clarity of the manuscript.

I have two major concerns – one technical, and one conceptual.

The technical concern concerns the procedure used to ‘control’ for innate attractiveness of a

patch, and thus for the non-biological explanations. The authors “assumed the first visit to each

2x2 mm pixel to be reflective of said properties” (line 318). I don't think this is a robust enough

assumption to hang a predictive claim on. I'm not saying there is a better option - I can't think of

one. But even if there is a correlation between the first pixel visit and some property of the pixel,

it will be so noisy as to be almost not there - purely descriptive. I don’t think it’s reasonable to

write unqualified claims about this in the abstract of the manuscript.

We agree with the sentiment and do no longer mention the second effect in the abstract.

My second, conceptual, issue, regards effect sizes. These are indeed reported, but not on any

useful, easily-comprehensible scale. I strongly advise providing comprehensible effect size

descriptions, such as “the top 10% most visited pixels had XXX% slower movement speeds and

XXX% lower sinuosity than the average pixel” or something similar. I will provide some further

examples in the detailed comments. Looking at the figures, it seems that the effect sizes are

uniformly very small. This (if it is the case) needs to be highlighted in the text, especially the

abstract and discussion, by the addition of adjectives such as “slightly”, “small”, and “minor”.

This strongly changes what readers will take away from the manuscript. In the conclusions, the

authors make some very important statements, which I think are otherwise hidden: that various

non-interactive effects (such as environmental idiosyncrasies) are likely to have an

overwhelmingly larger effect of movement patterns than any putative ‘home range marking

pheromone’ effect. I think this experiment shows that very nicely! But this barely comes out in

the abstract. Rather, the abstract focuses on, to me, possibly biologically unimportant effects,

while ignoring the more interesting message. All if this is hard to be certain of without some

clearer description of the effect sizes.

We added the suggested qualifying words to the abstract to convey the effect size information,

and replaced the mentioning of the “second” effect with that of the high sensitivity to

environmental features.

I also found the discussion, and indeed the whole paper, a bit on the long and rambling side.

However, this is a personal taste thing.

Overall, the study is well designed and well suited to filling a gap in the literature. I have no doubt

that it should be published, but am a bit concerned that the results do not strongly support all

the conclusions drawn.

MINOR COMMENTS

Title – “geometric artifacts” is not intuitive.

We changed the title to “Collective search in ants: Movement determines footprints, and

footprints influence movement”, following from a suggestion of Reviewer #1

Line 81 – I think a lot of readers might be confused when calling these CHCs pheromones. It’s a

reasonable term, but readers are used to thinking of trail or alarm pheromones in ants. Perhaps

be explicit first that you are terming them ‘pheromones’. Also somewhere it is important to

explicitly mention that this species does not use a recruitment pheromone.

We clarified the terminology and added a sentence on the lack of mass recruitment pheromone

use in the Study species section.

Line 127 5-20cm is a big range! Why such a range?

Good point: 15 cm away from one of the walls there was a strip of adhesive tape on the

underside of the uppermost paper layer, apparently acting as a ‘chemical wall’ to most ants,

inducing thigmotaxis. We thus changed this sentence to: “Points less than ca. 5 cm from the

walls or an apparently repellent tape strip on the underside of the top paper layer were

excluded,...”

Lines 130-132 – This was not easy for me to understand. Moreover, it was not clear why this

was done. This seems like a major alteration to the data, so the motivation should be clear.

“To avoid spurious angles resulting from tracking imprecision of still or stopping ants without

using arbitrary thresholding,...” was added to that sentence, as well as just following:

Resampling high-frequency movement tracks is an important step for analyzing the data on the

biologically most meaningful scale [Tourtellot et al. 1991, J Theor. Biol.]”.

Line 153 – reanalysis with 5 minutes and 5 hours – excellent! A really robust approach. I

applaud this.

Thank you!

Line 167 – was ‘ant’ included as a random effect, since ants provide multiple paths?

No, since we do not know which trajectory fragments belong to which ant.

Line 170 – Matlab, eh? Can this produce an analysis document? Like a knitted Rmarkdown file?

Would be good in the spirit of open science. I note that as the data was also provided as part of

the matlab file, I cannot access it or examine it, since I do not own Matlab. R is free and open

source, you know?

We regret that our choice of the analysis software hinders open science and will use better

alternatives in the future. The data are provided as .txt and we now also uploaded a markdown

file of the MATLAB script to the OSF folder.

Table 1 and elsewhere – “st~FP” is meaningless. Please use intelligible terms. I assume FP is

footprints? St is straightness?. Same for lines 250 and elsewhere

Correct, we straightened this out.

Line 201 – only a monster would separate the figure legends from the figures, and have the

figures hidden in the back. You’re not a monster, are you? I can only assume PLoS One insisted

on this. They, then, must be the monsters. No, but seriously, please next time keep the figures

with the legends in the main text.

WE are indeed not the monsters here and would like to second the appeal to PLoS to change

their guidance on this issue.

Lines 212-217 – this is an excellent paragraph; useful and clear. Can you please provide these

sort of summary statistics for the collective effects?

We added the % increase.

Line 251 – here for example adding something like “ants moved XXX% faster away from the

nest”. But throughout please illustrate the effect sizes.

We added easily interpretable numbers where possible.

Lines 294 – is this because most pixels only ever get walked over once? Maybe we are seeing

“slower more torturous ants spend longer at a location"?

There will only be one footprint counted per ant and pixel crossing, such that this correlation is

unlikely to explain that “...the less straight and the slower the first ant walked over that pixel, the

more footprints accumulated in the following time”.

As a response to your word choice, we must note that no ants in our experiments showed

abusive behavior towards other ants or the experimenters.

Line 311 – delete ‘to’

Done.

Line 392 – some sort of typo

Indeed, we forgot to finish this s. We completed it to say “We analyzed the data from the

‘cleaned’ trials as if there were still footprints present, to see if the pheromones are the cause of

the behavior changes.”

Lines 405 “…ants walk a tiny bit straighter and faster…” etc

Done.

Lines 496-500 – I agree completely, and feel like this should be a bigger message of the paper.

We hope that this is satisfactorily addressed by mentioning it in the abstract.

Lines 518 and thereafter – some citation issues going on here (2020)(5).

We included the year of the study in the text to distinguish it from the Hunt et al. 2016 study,

while the numbers in brackets are the ‘actual’ citation in the journal’s style.

Line 575 - small b

done.

---

## [Decision Letter · Decision Letter 2]

12 Feb 2024

Collective search in ants: Movement determines footprints, and footprints influence movement

PONE-D-23-30223R2

Dear Dr. Popp,

We’re pleased to inform you that your manuscript has been judged scientifically suitable for publication and will be formally accepted for publication once it meets all outstanding technical requirements.

**Kind regards,**

**Dr. Rahul Priyadarshi**

**Academic Editor**

**PLOS ONE**

---

## [Editor Report · Acceptance letter]

1 Mar 2024

PONE-D-23-30223R2 

PLOS ONE

Dear Dr. Popp, 

I'm pleased to inform you that your manuscript has been deemed suitable for publication in PLOS ONE. Congratulations! Your manuscript is now being handed over to our production team.

Kind regards, 

on behalf of

Dr. Rahul Priyadarshi 

Academic Editor

PLOS ONE